# SYNTHETIC CONTINUED PRETRAINING

**Zitong Yang**[*]
Department of Statistics
Stanford University

**Neil Band**[*]
Department of Computer Science
Stanford University

**Shuangping Li**
Department of Statistics
Stanford University

**Emmanuel Candès**
Department of Statistics
Stanford University

**Tatsunori Hashimoto**
Department of Computer Science
Stanford University

## ABSTRACT

Pretraining on large-scale, unstructured internet text enables language models to acquire a significant amount of world knowledge. However, this knowledge acquisition is *data-inefficient*—to learn a fact, models must be trained on hundreds to thousands of diverse representations of it. This poses a challenge when adapting a pretrained model to a small corpus of domain-specific documents, where each fact may appear rarely or only once. We propose to bridge this gap with *synthetic continued pretraining*: using the small domain-specific corpus to synthesize a large corpus more amenable to learning, and then performing continued pretraining on the synthesized corpus. We instantiate this proposal with EntiGraph, a synthetic data augmentation algorithm that extracts salient entities from the source corpus and then generates diverse text by drawing connections between those entities. Synthetic continued pretraining with EntiGraph enables a language model to answer questions and follow generic instructions related to the source documents without access to them. If the source documents are instead available at inference time, we show that the knowledge acquired through our approach compounds with retrieval-augmented generation. To better understand these results, we build a simple mathematical model of EntiGraph, and show how synthetic data augmentation can "rearrange" knowledge to enable more data-efficient learning.

## 1 INTRODUCTION

Language models (LMs) have demonstrated a remarkable ability to acquire knowledge from unstructured text, enabling them to perform challenging knowledge-intensive tasks (Brown et al., 2020; OpenAI et al., 2024; Gemini, 2024; Anthropic, 2024b; Dubey et al., 2024; Gunter et al., 2024). These successes are enabled by the combination of the next-token prediction objective (Shannon, 1951) and large-scale internet data (Common Crawl, 2007). However, it is becoming increasingly apparent that this approach is *data-inefficient;* for example, a 13-year-old human acquires knowledge from fewer than 100M tokens, while state-of-art open-source language models are trained on 15T tokens (Warstadt et al., 2023; Dubey et al., 2024). Recent works have highlighted a range of related problematic phenomena, including the "reversal curse", where models struggle to learn the relation "B=A" when trained on "A=B" (Berglund et al., 2023), and the requirement that models be exposed to thousands of examples per fact for knowledge acquisition (Allen-Zhu & Li, 2024).

These drawbacks pose a challenge when adapting the next-token prediction paradigm to learn from small-scale corpora. Because large-scale pretrained models already capture much of public common knowledge, further advancements will necessitate learning from the tails of the distribution (Kandpal et al., 2023): niche data that is either contained in small, private domains or appears only once or twice on the internet. This challenge of data-efficient, parametric knowledge acquisition is becoming increasingly important as growing compute capacity enables language model providers to exhaust publicly available data (Muennighoff et al., 2023; Villalobos et al., 2024).

We propose to address this problem of acquiring knowledge from small corpora with *synthetic continued pretraining*. To illustrate, consider the problem of teaching an LM a new area of mathematics, succinctly documented by a small set of textbooks. Directly training the model on those textbooks

---

[*]Equal contribution. Correspondence to: `zitong@berkeley.edu`, `nband@cs.stanford.edu`.

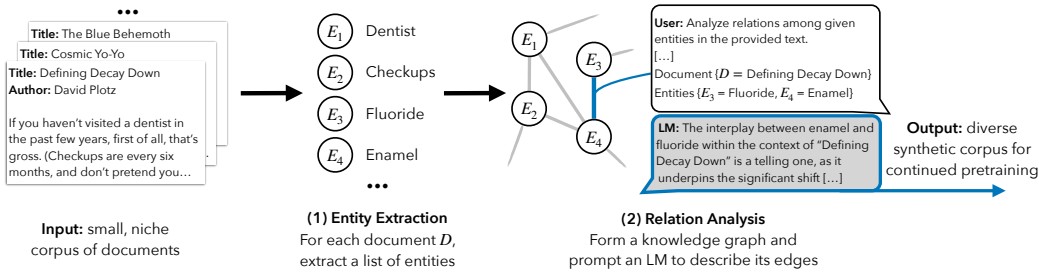

Figure 1: **Synthetic continued pretraining (synthetic CPT)** converts a small source corpus into a large synthetic corpus that is amenable to learning via standard continued pretraining. We instantiate synthetic CPT using a synthetic data augmentation algorithm called **EntiGraph**, which forms a knowledge graph over entities extracted from documents, and then prompts an LM to synthesize a text-based representation of the graph.

is unlikely to be effective due to the limited volume of text (e.g., tens of thousands of words), and the model will struggle to generalize from this compressed representation of knowledge. In contrast, learning established mathematical areas like linear algebra is straightforward because a large-scale corpus with diverse knowledge representations is accessible: for example, online lecture notes, Stack Exchange discussions, or Python implementations of the singular value decomposition. Synthetic continued pretraining bridges this gap by first converting a small, data-constrained domain into a synthetic corpus with diverse knowledge representations, and then continuing pretraining on it.

One basic approach is to simply paraphrase or rewrite the source documents in multiple ways. However, we demonstrate that this generic rephrasing does not cover the gap in the diversity of knowledge representations. We repeatedly rephrase a small corpus and find that the value of incremental synthetic data quickly decreases, with downstream model performance scaling poorly. We attribute this failure to the lack of diversity in paraphrasing alone. In the linear algebra example, online lecture notes and Stack Exchange discussions go beyond a simple rewrite of any textbook—they provide deeper analysis and application of the underlying concepts and techniques.

We address this shortcoming with EntiGraph, an entity-centric augmentation algorithm. EntiGraph breaks down a text corpus into a list of entities and then uses an LM to describe relations among entities, iteratively "filling in" the knowledge graph underlying the corpus (Figure 1).

To concretely measure progress towards effective knowledge acquisition from small corpora, we propose an experimental setting based on QuALITY (Pang et al., 2022), a reading comprehension dataset. It enables the evaluation of synthetic data generation methods for data-efficient learning without incurring the high compute costs of pretraining from scratch. Specifically, we assume access to a collection of 265 books totaling 1.3M tokens. Our task is to synthesize a corpus such that continued pretraining on it enables a model to answer queries (e.g., multiple-choice QA or user instructions related to the book content) *without* access to the source texts.

In our main experiments (§5), we use EntiGraph to generate 455M synthetic tokens from 1.3M real tokens using GPT-4 (OpenAI et al., 2024). Then, we continually pretrain Llama 3 8B (Dubey et al., 2024) on the synthetic tokens and evaluate its QA accuracy on the QuALITY questions. We observe log-linear scaling in the accuracy as synthetic token count increases, up to 455M (§4.2). At the endpoint, we find that synthetic continued pretraining with 455M EntiGraph tokens provides 80% of the accuracy gain of having the source documents available at inference time (§5). Beyond QA, we also perform instruction tuning on the continually pretrained model and find that it is capable of following open-ended instructions (e.g., summarization) related to the QuALITY books (§4.3).

To summarize, our key contributions are as follows:

- We propose to learn from small corpora with **synthetic continued pretraining**—converting the small corpus into a large, diverse, synthetic corpus and continuing pretraining on it—and instantiate this approach using the **EntiGraph** synthetic data augmentation algorithm (§2.2).
- We demonstrate that continued pretraining on the EntiGraph-synthesized corpus yields a QA accuracy scaling trend that is log-linear in the synthetic token count, significantly outperforming continued pretraining on the source documents or paraphrases (§4.2). Furthermore, we show that

instruction tuning the EntiGraph continually pretrained model enables it to follow more diverse queries related to the source documents (§4.3).

- We complement the main experiments with an open-book setup (§5), providing the model with access to the source documents when answering queries. We demonstrate that the knowledge acquired through synthetic continued pretraining with EntiGraph is *complementary* to the knowledge accessed through retrieval-augmented generation (RAG, Lewis et al. (2020))—RAG with the EntiGraph continually pretrained model outperforms RAG with the base model.
- Lastly, we build a mathematical model that captures the intuition behind EntiGraph. We analyze it to obtain a parametric formula for the scaling trend of a continually pretrained model's accuracy with respect to EntiGraph synthetic tokens, closely matching our empirical observations (§6).

Practically, synthetic continued pretraining with EntiGraph enables pretrained LMs to adapt to specialized domains by acquiring *parametric* knowledge, rather than the non-parametric knowledge accessed through retrieval. At a higher level, our approach points toward a family of synthetic data generation algorithms that convert compute into data efficiency for (continued) pretraining.

## 1.1 RELATED WORK

We next discuss recent work most related to our setting of synthetic data generation for continued pretraining. Appendix B surveys classical work on synthetic data and continual learning.

**Synthetic generation of pretraining data.** Recent approaches synthesize *pretraining* data using hierarchical prompting methods to promote dataset diversity. Eldan & Li (2023) prompt LLMs to generate stories containing sampled keywords, and demonstrate that small LMs trained on their dataset can generate fluent text. Gunasekar et al. (2023) synthesize textbooks and code exercises by conditioning on topic, target audience, and function names, and later release strong LLMs pretrained on synthetic data (Li et al., 2023b; Abdin et al., 2023; 2024). However, their datasets and prompts are not public. Maini et al. (2024) prompt an LM to rephrase documents for pretraining, improving training efficiency. Distinct from all above works, our focus is teaching a pretrained LLM the knowledge of a small corpus. Mecklenburg et al. (2024) consider task-specific finetuning and propose a fact-based synthetic QA generation procedure, but do not show improvement on generic instruction following tasks. We instead focus on teaching a model generally useful knowledge about a small corpus, untied to a particular downstream task. Ovadia et al. (2024) continually pretrain Llama 2–based LMs on synthetic paraphrases of Wikipedia articles, but do not observe consistent improvements. We adapt the approach of Maini et al. (2024) and Mecklenburg et al. (2024) to our small corpus setting ("Rephrase baseline" in §4). We find that our graph-based augmentation algorithm outperforms it, likely because our approach enforces diversity through entity-based generation.

**Continued pretraining.** Continual or continued *pretraining* works (Gururangan et al., 2020) adapt pretrained LLMs to broad target domains such as code, medicine, or mathematics by collecting massive datasets (often >100B tokens; cf. Table 1 for a survey) and applying causal language modeling recipes (Gupta et al., 2023; Ibrahim et al., 2024; Parmar et al., 2024). We aim to extend the success of continued pretraining to small, specialized domains such as proprietary datastores. Observing that standard continued pretraining is ineffective on small corpora, we propose a knowledge graph–inspired approach to synthesize a diverse related corpus and find it more amenable to learning.

**Knowledge editing.** A related line of work updates LMs with small units of factual knowledge, e.g., (subject, relation, object) tuples. Zhu et al. (2020) study constrained fine-tuning to limit model complexity. Later approaches attempt to localize where factual knowledge is stored in Transformers and update only those weights (Mitchell et al., 2022; Meng et al., 2022; 2023), or maintain an external memory of edits and prepend them as context during generation (Zhong et al., 2023; Cohen et al., 2023). Most related to our work is Akyürek et al. (2024), which first deduces implications of a factual edit and then finetunes on those implications. Unlike the knowledge editing literature which learns atomic, sentence-length facts, we aim to learn from a small corpus of documents.

## 2 OUR METHOD

We focus on learning parametric knowledge from a small corpus of documents. Our goal is to continually pretrain an LM to acquire the knowledge of a niche corpus. Observing that simple continued pretraining is ineffective (§4), we propose to use synthetic continued pretraining, which first uses the small corpus to synthesize a larger one more amenable to learning, and then continues

| Study | Domain | Model Parameter Count | Total Unique CPT Tokens |
|---|---|---|---|
| Minerva (Lewkowycz et al., 2022) | STEM | 8B, 62B, 540B | 26B-38.5B |
| MediTron (Chen et al., 2023) | Medicine | 7B, 70B | 46.7B |
| Code Llama (Rozière et al., 2024) | Code | 7B, 13B, 34B | 520B-620B |
| Llemma (Azerbayev et al., 2024) | Math | 7B, 34B | 50B-55B |
| DeepSeekMath (Shao et al., 2024) | Math | 7B | 500B |
| SaulLM-7B (Colombo et al., 2024b) | Law | 7B | 30B |
| SaulLM-{54, 141}B (Colombo et al., 2024a) | Law | 54B, 141B | 520B |
| HEAL (Yuan et al., 2024a) | Medicine | 13B | 14.9B |
| Our setting | Articles & Books | 8B | 1.3M |

Table 1: Comparing the scale of modern continued pretraining (CPT) works with our small corpus setting. Prior work adapts LMs to broad domains with diverse, large-scale corpora. We aim to downscale CPT to small corpora; we use a corpus that is $10{,}000\times$ smaller than the smallest modern corpus for domain-adaptive CPT.

pretraining on the synthetic corpus. In this section, we first outline this problem setting and our evaluation approach in more detail (§2.1). Then, we provide a concrete instantiation of synthetic continued pretraining using a data augmentation algorithm called EntiGraph (§2.2).

## 2.1 PROBLEM SETUP

**Continued pretraining on small corpora.** We focus on approaches that continually pretrain an LM to teach it the knowledge of a small source corpus $\mathcal{D}_{\text{source}}$. These approaches acquire "parametric knowledge"—the knowledge of $\mathcal{D}_{\text{source}}$ is learned in the LM's parameters, as in pretraining.

**Synthetic continued pretraining (synthetic CPT).** First, we apply a synthetic data generation algorithm $\mathcal{A}_{\text{synth}}$ to convert a small corpus $\mathcal{D}_{\text{source}}$ into a synthetic corpus $\mathcal{D}_{\text{synth}}$:

$$\mathcal{A}_{\text{synth}} : \mathcal{D}_{\text{source}} \longmapsto \mathcal{D}_{\text{synth}}. \tag{1}$$

Then, we perform continued pretraining on $\mathcal{D}_{\text{synth}}$ instead of on $\mathcal{D}_{\text{source}}$. We implement $\mathcal{A}_{\text{synth}}$ using a prompted LM. A natural concern is that the LM may hallucinate and fabricate false knowledge. Therefore, we consider **synthetic data augmentation** algorithms that condition the generation process on the source documents to improve the synthesized data's faithfulness.

**Evaluation with knowledge-intensive queries.** We evaluate the quality of a synthetic data augmentation algorithm $\mathcal{A}_{\text{synth}}$ by testing whether the downstream synthetic CPT model has effectively acquired the knowledge of $\mathcal{D}_{\text{source}}$ in its parameters. More precisely, we curate test queries $\mathcal{Q}_{\text{test}}$ that probe the knowledge about $\mathcal{D}_{\text{source}}$ acquired by the model. For example, in the linear algebra setting, $\mathcal{Q}_{\text{test}}$ could be held-out exam questions. To test parametric knowledge, we do not allow the model to access the source documents $\mathcal{D}_{\text{source}}$ at test time. Therefore, the queries cannot be ambiguous without access to $\mathcal{D}_{\text{source}}$. For example, a reading comprehension question like "Where was he born?" is ambiguous without context. Altogether, we can evaluate data augmentation algorithms $\mathcal{A}_{\text{synth}}$ for synthetic CPT using a paired source corpus and related test queries $(\mathcal{D}_{\text{source}}, \mathcal{Q}_{\text{test}})$.

## 2.2 ENTIGRAPH

Next, we present EntiGraph, our instantiation of a synthetic data augmentation algorithm $\mathcal{A}_{\text{synth}}$. At a high level, EntiGraph generates diverse representations of knowledge from a small corpus $\mathcal{D}_{\text{source}}$ by using a prompted LLM to synthesize a knowledge graph representation of $\mathcal{D}_{\text{source}}$. EntiGraph consists of two steps/prompts: extracting entities from the document and analyzing relations among an arbitrary subset of the entities (Figure 1). Altogether, this hierarchical prompting strategy *externalizes* the problem of generating diverse synthetic text to a combinatorial structure—namely, a graph relating various entities appearing in the corpus documents.

**Step 1: Entity extraction.** First, EntiGraph extracts a list of salient entities $\{E_1, E_2, \ldots, E_n\}$ from the document $\mathcal{D}_{\text{source}}$ using an `entity_extraction` prompt (full prompt in Appendix H.1): $\{E_1, E_2, \ldots, E_n\} \sim \text{LM}_{\text{aug}}\big(\text{entity\_extraction}(\mathcal{D}_{\text{source}})\big)$. In the linear algebra example, $\mathcal{D}_{\text{source}}$ could be one specific linear algebra textbook. We would expect to extract entities such as $\{E_1 = \text{Linear space}, E_2 = \text{Vector}, E_3 = \text{SVD}, \ldots\}$.

**Step 2: Relation analysis.** Next, EntiGraph analyzes the relations among subsets of entities. The intuition is to explore the edges of the knowledge graph underlying the source document $\mathcal{D}_{\text{source}}$, analogous to a student writing diverse notes about a linear algebra textbook. We apply a `relation_analysis` prompt (full prompt in Appendix H.1) to describe how a sub-

set of $k \leq n$ entities are related in the context of the source document $\mathcal{D}_{\text{source}}$: $\widetilde{D}_{E_{i_1} \ldots E_{i_k}} \sim$ $\mathsf{LM}_{\text{aug}}\big(\texttt{relation\_analysis}(D, E_{i_1}, E_{i_2}, \ldots, E_{i_k})\big)$. For example, if $E_1 = \texttt{Linear space}$ and $E_2 = \texttt{Vector}$, $\widetilde{D}_{E_1 E_2}$ could be $\texttt{Based on the textbook, a vector is an}$ $\texttt{element of a linear space...}$ Exhaustively enumerating all possible subsets of entities is impractical. We generate data for pairs $\widetilde{D}_{E_i E_j}$ and triplets $\widetilde{D}_{E_i E_j E_k}$ in our experiments.

**EntiGraph synthetic corpora.** Finally, we collect all sampled synthetic texts from Step 2 as the EntiGraph output: $\mathcal{D}_{\text{EntiGraph}} = \{\widetilde{D}_{E_{i_1} \ldots E_{i_k}}, \ldots\}$. Altogether, we described a data augmentation algorithm mapping a small source corpus $\mathcal{D}_{\text{source}}$ to a larger synthetic corpus $\mathcal{D}_{\text{EntiGraph}}$, as in (1).

## 3 EXPERIMENT SETUP

We next detail how we evaluate a given data augmentation algorithm $\mathcal{A}_{\text{synth}}$. As described in §2.1, we evaluate algorithms $\mathcal{A}_{\text{synth}}$ by evaluating how well an LM continually pretrained on their output synthetic corpus $\mathcal{A}_{\text{synth}}(\mathcal{D}_{\text{source}})$ can answer test queries $\mathcal{Q}_{\text{test}}$ about the source documents $\mathcal{D}_{\text{source}}$.

In our main experiments, we use queries that are unambiguous without the source documents $\mathcal{D}_{\text{source}}$, and disallow the LM from accessing $\mathcal{D}_{\text{source}}$ while answering queries $\mathcal{Q}_{\text{test}}$. This allows us to evaluate which data augmentation algorithm best promotes the acquisition of parametric knowledge through synthetic CPT. Later, in §5, we consider an open-book setting where the model can simultaneously access the source documents $\mathcal{D}_{\text{source}}$ and test queries $\mathcal{Q}_{\text{test}}$, to test how the parametric knowledge acquired through synthetic CPT composes with non-parametric access to knowledge through retrieval (Lewis et al., 2020). We next introduce our small corpus and related test queries $(\mathcal{D}_{\text{source}}, \mathcal{Q}_{\text{test}})$.

**QuALITY corpus $\mathcal{D}_{\text{source}}$.** Our corpus and test queries are based on the QuALITY (Pang et al., 2022) long-document comprehension benchmark. The QuALITY corpus $\mathcal{D}_{\text{source}}$ consists of 265 articles and short books on genres such as science fiction and journalism, averaging ~5,000 tokens.

**QuALITY test queries $\mathcal{Q}_{\text{test}}$.** We use the 10-20 multiple choice questions accompanying each article in QuALITY. They serve as high-quality knowledge probes on $\mathcal{D}_{\text{source}}$, but the query phrasing often presupposes the reading comprehension context (e.g., "What does the author think about..."). We remove ambiguity by contextualizing them with an article reference: "In the context of article {article_name} by {author_name}, what does the author think about...". This provides us with 4,609 unambiguous queries $\mathcal{Q}_{\text{test}}$ to test the parametric knowledge of our continually pretrained LMs.

**Evaluation on instruction-tuned summarization.** We also instruction tune the continually pretrained LMs and evaluate them on more general instruction following queries. Specifically, we prompt them to generate closed-book summaries of QuALITY articles, given only title and author.

**Performance with strong API-based LLMs.** In our continued pretraining setting, we must select a corpus $\mathcal{D}_{\text{source}}$ that is not well-represented in standard pretraining datasets. As an initial test of the obscurity of the QuALITY corpus $\mathcal{D}_{\text{source}}$, we evaluate GPT-3.5 and GPT-4 on $\mathcal{Q}_{\text{test}}$. In the closed-book setting, we find GPT-3.5 accuracy at 44.81% and GPT-4 accuracy at 51.30% (Figure 2). In the open-book setting (full access to $\mathcal{D}_{\text{source}}$), we find GPT-3.5 accuracy at 72.60% and GPT-4 accuracy at 86.09% (Table 3). Based on the large (~30%) improvement when $\mathcal{D}_{\text{source}}$ is provided, we conclude that the QuALITY corpus $\mathcal{D}_{\text{source}}$ is sufficiently niche to serve as an appropriate testbed.

## 4 MAIN EXPERIMENTS

In this section, we present our main experimental results[1]. Using GPT-4[2] as our prompted model $\mathsf{LM}_{\text{aug}}$, we apply EntiGraph to the 1.3M token QuALITY corpus $\mathcal{D}_{\text{source}}$, generating a 455M token synthetic corpus. For the remainder of the paper, we refer to the former as the "Raw corpus" and the latter as the "EntiGraph corpus". Additional details on these corpora are provided in Appendix C.

We continually pretrain Llama 3 8B (Dubey et al., 2024) with causal language modeling on the 455M token EntiGraph corpus. In §4.1, we describe our CPT procedure and introduce two natural baselines. In §4.2, we evaluate on the QuALITY test queries $\mathcal{Q}_{\text{test}}$. In §4.3, we show that synthetic CPT using EntiGraph is compatible with downstream instruction tuning (Ouyang et al., 2022).

---

[1]Code https://github.com/ZitongYang/Synthetic_Continued_Pretraining.git.
[2]We use the gpt-4-turbo model as of Aug. 19, 2024.

## 4.1 CONTINUED PRETRAINING PROCEDURE

**EntiGraph CPT.**  In our main continued pretraining experiment, we continually pretrain Llama 3 8B Base on the 455M token EntiGraph corpus for 2 epochs with replay on the RedPajama dataset (TogetherAI, 2023). Hereafter, we refer to this model as "EntiGraph CPT". We discuss CPT details in Appendix D. Next, we describe two baselines to which we compare in closed-book QA (§4.2).

**Raw CPT baseline.**  The first baseline continues pretraining Llama 3 8B Base on the 1.3M token Raw corpus of raw QuALITY articles $\mathcal{D}_{source}$. We jointly tune the number of epochs and RedPajama replay rate, obtaining the "Raw CPT" model. Further tuning details are provided in Appendix D.

**Rephrase CPT baseline.**  Another simple synthetic data augmentation procedure is to rephrase QuALITY articles repeatedly. Maini et al. (2024) and Ovadia et al. (2024) execute a systematic extension of this idea (cf. §1.1). Based on their approaches, we craft a "Rephrase baseline" which repeatedly applies three fixed prompts (easy, medium, and hard rephrase)[3] to the QuALITY articles at temperature 1.0. We stopped generating paraphrases at 38M tokens, where we observed a clear gap in QA evaluations from EntiGraph CPT and a slower scaling trend (Figure 2). We refer to this data as the "Rephrase corpus" and the continually pretrained model as "Rephrase CPT".

## 4.2 QUESTION-ANSWERING EVALUATIONS

Next, we present our closed-book QA evaluations with the QuALITY test queries $\mathcal{Q}_{test}$.

**Evaluation procedure.**  Each QuALITY question is a four-choice, single-answer multiple choice question (similar to MMLU, Hendrycks et al. (2021)). We evaluate with 5-shot chain-of-thought prompting (Brown et al., 2020; Wei et al., 2024) and provide our prompt in Appendix I.1.

**EntiGraph scaling.**  We find that CPT on the 455M token EntiGraph corpus improves closed-book QA accuracy from 39.49% (for Llama 3 8B Base) to 56.22% (Figure 2). A natural question is how accuracy scales as we synthesize and train on more tokens with Enti-Graph. To test this, we randomly subsample without replacement the EntiGraph corpus with varying sample sizes, continually pretrain Llama 3 8B Base on each subsample, and plot accuracy versus sample size in Figure 2. We observe log-linear scaling of the accuracy in the number of synthetic tokens used for CPT, up to 455M tokens. We mathematically investigate the scaling properties of EntiGraph in §6. In broad strokes, we postulate that QuALITY accuracy follows a mixture-of-exponential shape with three stages: (i) linear growth, (ii) log-linear growth, and (iii) asymptotic plateau.

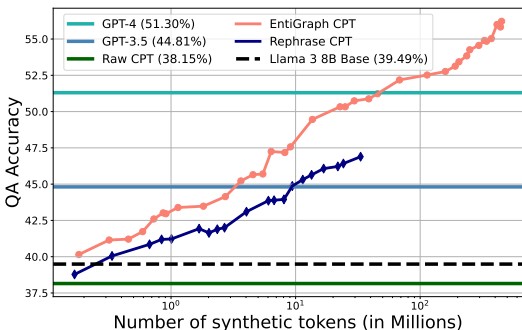

Figure 2: Accuracy on the QuALITY question set $\mathcal{Q}_{test}$ ($y$-axis) as a function of the synthetic token count ($x$-axis). The accuracy of synthetic continued pretraining using the EntiGraph data augmentation algorithm (Enti-Graph CPT) scales log-linearly up to 455M tokens.

**Comparison with baselines.**  Raw CPT (green line) underperforms even Llama 3 8B (dashed black line). We postulate two explanations: (i) The Raw corpus follows a narrower, different distribution than the Llama 3 pretraining corpus; heavily training on it may harm the LM's English capabilities. (ii) The limited diversity of knowledge representations in the Raw corpus leads to limited knowledge acquisition due to problems such as the reversal curse (Berglund et al., 2023). Rephrase CPT scales poorly compared with EntiGraph (Figure 2), suggesting that for synthetic CPT to scale well, the synthetic data must be sufficiently diverse. EntiGraph tackles this problem using a hierarchical prompting strategy that externalizes diversity to a knowledge graph's combinatorial relationships.

## 4.3 INSTRUCTION FOLLOWING EVALUATIONS

Next, we explore more general test queries beyond the test queries $\mathcal{Q}_{test}$. Concretely, we perform instruction tuning on EntiGraph CPT to obtain EntiGraph Instruct. We demonstrate that synthetic CPT on the EntiGraph corpus is compatible with instruction tuning; EntiGraph Instruct can directly use

---

[3]Maini et al. (2024) include a 4th prompt to generate synthetic QA pairs. We defer this task-specific QA finetuning method to Appendix E and focus on task-agnostic baselines for learning generic knowledge.

knowledge obtained during synthetic CPT in instruction following tasks, without test-time access to the QuALITY corpus $\mathcal{D}_{\text{source}}$. We detail our instruction tuning procedure in Appendix D.

**Instruction tuning qualitative examples.** We first present qualitative examples that demonstrate EntiGraph Instruct's ability to follow instructions related to QuALITY articles. First, we ask the model to summarize a QuALITY article with explicit reference to the title and author, but no access to the article itself (Table 2, top row). Next, we show that even without an explicit reference to the title and author, knowledge of the article is stored in the model's parameters and can affect its behavior (Table 2, middle row). Finally, we provide an example where the model performs a comparison using knowledge across two articles (Table 2, bottom row). Albeit artificial, this shows that though EntiGraph does not synthesize data that simultaneously involves multiple articles, the model can reason about their interaction using its parametric knowledge. We provide full responses in Table 6.

**Evaluating closed-book summarization.** We also present quantitative metrics for summarization, a well-studied instruction following task. We compare EntiGraph Instruct summaries of QuALITY articles with human-written summaries from sQuALITY (Wang et al., 2022), a variation of QuALITY with provided human summaries. Common scalar summarization metrics such as ROUGE (Lin, 2004) or BERTScore (Zhang* et al., 2020) mostly evaluate text similarity between the summary and source articles, and may not accurately reflect summarization quality for abstractive systems (Zhang et al., 2024b). We use a simple, automated evaluation metric based on pyramid evaluation (Nenkova et al., 2007; Gao et al., 2019) that measures both the hallucination rate and how well the summary captures the salient claims of the original article. Our approach uses GPT-4 to (1) split the summary into atomic claims (Min et al., 2023), (2) decide whether each claim is true/false based on the source article, and (3) determine if true claims are salient to the article's main message. We hence obtain the count of false and salient claims for each summary, normalize these by the corresponding count from the human summary, and report the average of these normalized metrics in Figure 3. Appendix I.2 provides further details.

Table 2: EntiGraph Instruct examples.

| |
|---|
| **Explicit reference:** Summarize "Defining Decay Down". |
| The article "Defining Decay Down" by David Plotz discusses [...] Dentists began to focus on cosmetic dentistry, [...] |
| **Implicit reference:** How has dentistry in the U.S. changed? |
| 1. Increase in cosmetic dentistry [...] 
 2. Use of technology: [...] |
| **Cross article instruction:** Compare David Plotz's commentary on American dentistry and the movie Fight Club? |
| David Plotz's commentary style is different when he analyzes American dentistry and when he discusses the movie Fight Club. [...] |

**Results discussion.** In Figure 3, we compare four summarizers: EntiGraph Instruct, Raw Instruct, GPT-3.5, and GPT-4. We provide each summarizer with two different prompts, asking for short and long summaries (prompts in Appendix I.2). When we request more detailed summaries, Raw Instruct hallucinates and generates more false claims with little improvement in the number of salient claims. In contrast, EntiGraph Instruct can generate more salient claims as the summary gets longer, with a small increase in the number of false claims (similar to GPT-3.5 and GPT-4 levels). The gaps in both salient and false claim rates are sufficiently large that these results likely hold beyond our particular metric. We complement the automated evaluation metrics above with several qualitative examples in Appendix I.2.

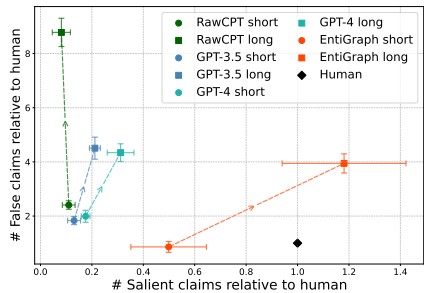

Figure 3: Closed-book summarization: number of false claims ($y$-axis) versus number of salient claims ($x$-axis) normalized by the human summary.

## 5 OPEN-BOOK EXPERIMENTS

Next, we consider an open-book setting with the domain-specific corpus $\mathcal{D}_{\text{source}}$ available at test time. In this widespread setting, retrieval-augmented generation (RAG; Lewis et al. (2020)) is the predominant approach. A natural question whether the parametric knowledge learned through synthetic CPT using EntiGraph complements the non-parametric knowledge accessed using RAG. We answer this question by comparing a state-of-the-art RAG pipeline with and without Entigraph CPT.

| EntiGraph CPT + RAG | | Llama 3 8B Base + RAG | | GPT-4 + Oracle RAG | | GPT-3.5 + Oracle RAG | |
|---|---|---|---|---|---|---|---|
| Accuracy | Recall@8 | Accuracy | Recall@8 | Accuracy | Recall@8 | Accuracy | Recall@8 |
| 62.60 | 99.63 | 60.35 | 99.63 | 86.09 | 100.0 | 72.60 | 100.0 |

Table 3: QuALITY question-answering accuracy and recall rate in the open-book retrieval-augmented generation (RAG) setting. EntiGraph CPT and Llama 3 8B Base are used in a RAG pipeline (cf. §5 for setup details). Recall@8 is defined as the proportion of questions for which the salient article appears in the top 8 reranked document chunks. GPT-4 and GPT-3.5 Oracle RAG provide an upper bound with a perfect retriever, by placing the entire relevant document in-context.

**RAG evaluation setup.** Our RAG pipeline follows established best practices (Lewis et al., 2020; Gao et al., 2024). It involves an offline stage which indexes document chunks, followed by inference-time retrieval, reranking, and placement of those chunks in a few-shot LM prompt. Throughout, we use OpenAI `text-embedding-3-large` (Neelakantan et al., 2022) as our API-based embedding model, FAISS as our similarity search index (Douze et al., 2024), and Cohere `rerank-english-v3.0` (Cohere, 2024) as our reranker. Following the evaluation procedure detailed in §4, we evaluate parallel RAG pipelines on the QuALITY multiple choice test set using few-shot chain-of-thought prompting. All hyperparameters are tuned separately for each LM's RAG pipeline. We refer the reader to Appendix F for further details on our RAG evaluation setup.

**EntiGraph continued pretraining complements RAG.** We observe in Table 3 that EntiGraph CPT outperforms Llama 3 8B Base, the model from which it is continually pretrained. These results demonstrate that the knowledge internalized through synthetic CPT is complementary to that accessed during RAG, and demonstrate a competitive new recipe for small corpus QA: (1) synthetic data augmentation, (2) continued pretraining, and (3) RAG.

**EntiGraph continued pretraining alone approaches RAG performance.** These results also contextualize the effectiveness of EntiGraph in the closed-book, parametric knowledge setting (§4). Comparing Figure 2 and Table 3, we observe that adding RAG to Llama 3 8B Base improves accuracy by $20.86\%$ ($39.49\% \rightarrow 60.35\%$). On the other hand, continued pretraining of Llama 3 8B Base on the EntiGraph corpus improves accuracy by $16.73\%$ ($39.49\% \rightarrow 56.22\%$). Hence, EntiGraph continued pretraining provides $> 80\%$ of the absolute performance improvement of RAG, even in a small corpus setting where RAG recall is nearly perfect.

Overall, our results show that the parametric knowledge acquired in EntiGraph continued pretraining composes with realistic knowledge-intensive QA pipelines, and that EntiGraph continued pretraining alone—without test-time corpus access—is nearly competitive with a strong RAG baseline.

## 6 THEORETICAL ANALYSIS OF ENTIGRAPH SCALING

It may seem surprising that simply "rewriting" the source documents $\mathcal{D}_{\text{source}}$ can improve performance at all (§4), as EntiGraph does not explicitly add new knowledge beyond $\mathcal{D}_{\text{source}}$. We postulate that EntiGraph "rearranges" $\mathcal{D}_{\text{source}}$ into a layout more amenable to learning. For example, in $\mathcal{D}_{\text{source}}$, the entity pair $(A, B)$ may appear together in some sentences and $(B, C)$ in others. As a result, models trained directly on $\mathcal{D}_{\text{source}}$ may learn the $(A, B)$ relation and the $(B, C)$ relation, but not the $(A, C)$ relation (Akyürek et al., 2024). We build a mathematical model to formalize this intuition (§6.1) and provide a quantitative prediction that the scaling trend of EntiGraph CPT follows a mixture-of-exponential shape (§6.3), which fits well with our empirical observations (Figure 4).

### 6.1 TOY MODEL SETUP

In this toy model, we use $\mathcal{V}$ to denote the set of entities, and represent the source documents $\mathcal{D}_{\text{source}}$ with pairs of known relations $\mathcal{D}_{\text{source}} \subset \{(x, y) \in \mathcal{V}^2 : x \neq y\}$. We assume that each relation pair in $\mathcal{V}^2$ appears in the source documents $\mathcal{D}_{\text{source}}$ independently at random, with probability $p$. Mathematically, $\mathbb{P}[(x, y) \in \mathcal{D}_{\text{source}}] = p$ for all $x \in \mathcal{V}$ and $y \in \mathcal{V}$ with $x \neq y$. We write $V = |\mathcal{V}|$ and assume that $p = \lambda/V$, for some constant $\lambda > 1$.

**Training as memorization.** We model the learning of factual knowledge as a memorization process, in which a model memorizes the relations it is trained on but does not meaningfully generalize beyond them (Yang et al., 2023; Feldman, 2020). In this view, a language model's knowledge can be represented by a matrix $\boldsymbol{M} \in \{0, 1\}^{V \times V}$ such that $\boldsymbol{M}(x, y) = 1$ if the model "knows" the $(x, y)$ relation and equals 0 otherwise. Then, training directly on the source documents $\mathcal{D}_{\text{source}}$ sim-

ply means setting all entries that appear in $\mathcal{D}_{\text{source}}$ to 1, denoting that the model has memorized the relations given in the source documents. Mathematically, we denote this model trained on $\mathcal{D}_{\text{source}}$ by the matrix $\boldsymbol{M}_0 \in \{0,1\}^{V \times V}$, which has i.i.d. Bernoulli off-diagonal entries with mean $p$.

**EntiGraph synthetic data augmentation.** Given the source documents $\mathcal{D}_{\text{source}}$, we define the following iterative procedure of synthetic data generation: for each $t = 1, 2, \ldots$

- **Entity pair selection:** Sample $(x_t, y_t) \in \{(x,y) \in \mathcal{V}^2 : x \neq y\}$ uniformly at random.
- **Relation analysis:** Generate the "relation between $(x_t, y_t)$" by performing a breadth-first search (BFS) on the directed graph represented by the adjacency matrix $\boldsymbol{M}_0$ starting at $x_t$. If no such path exists, do nothing. If there exists a path $(x_t, z_t^1, z_t^2, \ldots, z_t^{k_t}, y_t)$ connecting $x_t$ to $y_t$, define $\mathcal{D}_t = \{(x_t, z_t^1), (x_t, z_t^2), \ldots, (x_t, z_t^{k_t}), (x_t, y_t)\} \cup \mathcal{D}_{t-1}$, where we assume $\mathcal{D}_0 = \mathcal{D}_{\text{source}}$. The model trained on this round of synthetic data is $\boldsymbol{M}_t = \boldsymbol{M}_{t-1} + \sum_{(x,y) \in \mathcal{D}_t \setminus \mathcal{D}_{t-1}} \boldsymbol{I}_{xy}$, where $\boldsymbol{I}_{xy} \in \{0,1\}^{V \times V}$ is a binary matrix with $\boldsymbol{I}_{xy}(x,y) = 1$ and 0 otherwise.

This mirrors the relation analysis step for the EntiGraph synthetic data augmentation algorithm (Step 2, §2.2). With the setup above, the index $t$ is analogous to the number of synthetic tokens that the model has generated, and the model's knowledge is captured by how many ones the matrix $\boldsymbol{M}_t$ contains. To make this connection precise, we define the link density (or accuracy) of $\boldsymbol{M}_t$ to be $\mathsf{Acc}(\boldsymbol{M}_t) = \mathbb{E}[\|\boldsymbol{M}_t\|_1|\boldsymbol{M}_0]/(V(V-1))$, where the expectation is taken over the randomness arising from the synthetic data generation process and not the source documents $\mathcal{D}_{\text{source}}$, and $\|M\|_1$ denotes $\sum_{i,j} |M_{i,j}|$. We use the notation $\mathsf{Acc}$ as this is intended to emulate the accuracy on QuALITY test queries studied in the experimental sections (§4 and §5).

## 6.2 RIGOROUS UPPER AND LOWER BOUND

In this section, we derive rigorous upper and lower bounds on the scaling trend of $\mathsf{Acc}(\boldsymbol{M}_t)$.

**Definition 1.** *Let $C_\lambda = (1 - \rho(\lambda))^2$, where $\rho(\lambda)$ denotes the extinction probability for a Poisson$(\lambda)$ branching process (i.e., $\rho$ is the smallest solution in $[0,1]$ to the fixed-point equation $\rho = \exp(\lambda(\rho - 1))$). For any fixed $\varepsilon > 0$, we further define $C_{\text{LB}} = 1 - \frac{1}{V(V-1)}, \quad C_{\text{UB}} = 1 - \frac{(1+\varepsilon)\log V}{V(V-1)\log \lambda}$.*

**Theorem 1.** *For any time $t \geq 1$ and any $\varepsilon > 0$, the link density satisfies, with probability $\to 1$,*

$$\left(p + C_\lambda \left(1 - C_{\text{LB}}^t\right)\right)(1 - \varepsilon) \leq \mathsf{Acc}(\boldsymbol{M}_t) \leq \left(p + C_\lambda \left(1 - C_{\text{UB}}^t\right)\right)(1 + \varepsilon) \text{ as } V \to \infty.$$

Even though Theorem 1 provides mathematically rigorous upper and lower bounds on the scaling trend of $\mathsf{Acc}(\boldsymbol{M}_t)$, the exact growth curve is more intricate, as we will show next.

## 6.3 AN ANALYTICAL FORMULA

We analyze the link density $\mathsf{Acc}(\boldsymbol{M}_t)$ using a Poisson branching process approximation of the cluster growth of vertices. This approach yields a *mixture-of-exponential* scaling trend

$$\mathsf{Acc}(\boldsymbol{M}_t) \sim p + C \left(1 - \sum_{k=1}^{\infty} \mu(k) (1 - a_k)^t\right), \quad (2)$$

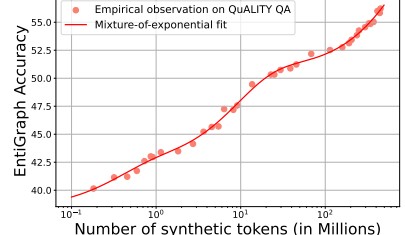

Figure 4: A mixture-of-exponential function (2) closely fits the scaling trend of EntiGraph CPT with respect to synthetic token count.

where $A \sim B$ means that $A/B$ converges to 1 in probability as $V \to \infty$. The parameter $C$ governs the link density $\mathsf{Acc}(\boldsymbol{M}_t)$ as $t \to \infty$ and is determined by the proportion of reachable pairs of vertices in the initial matrix $\boldsymbol{M}_0$. $\mu(\cdot)$ is the probability mass function on $k$, which controls the proportion of pairs of vertices with a specific decay rate. The parameters $\mu(\cdot)$ and $a_k$ depend on $\boldsymbol{M}_0$ in a more intricate manner (cf. Appendix G for a full derivation). We find that (2) accurately fits the empirical scaling trend of EntiGraph CPT accuracy up to 455M synthetic tokens (Figure 4). We discuss curve fitting in Appendix G.1, where we show that the mixture-of-exponential shape grows in three phases: (i) linear growth; (ii) log-linear growth; (iii) asymptotic plateau.

## 7 DISCUSSION AND CONCLUSION

### 7.1 LIMITATIONS

Because EntiGraph synthesizes data using a prompted LM, there is a risk it may hallucinate and fabricate non-existent entities or relations. Although our synthesis process is grounded by the source

documents, it is an assumption that $LM_{aug}$ is capable enough to generate faithful synthetic data when conditioned on $\mathcal{D}_{source}$. We quantitatively test the factuality of the EntiGraph corpus by randomly subsampling 150 sentences from it and manually labeling each sentence's factuality (Appendix A.2). We find roughly half of the sentences are subjective, and the other objective half is almost always factual. We postulate that factuality is high because QuALITY articles are relatively simple given the capability of the prompted LM. If EntiGraph were applied to more challenging content like a complex research paper, it is possible that the prompted model could be more prone to hallucination.

On the other hand, since we use a strong prompted LM `gpt-4-turbo` to generate synthetic data, one might be concerned that our performance gains come from distilling it. To probe this, we perform an ablation study where we replace `gpt-4-turbo` with Llama 3.1 8B Instruct, a substantially weaker model that is obtained from the same base model as EntiGraph CPT, in Appendix A.1. We generated 334M EntiGraph tokens using Llama 3.1 8B Instruct and found a consistent log-linear trend with the same slope but lower intercept (Figure 5) compared with GPT-4 generation. This ablation suggests that EntiGraph operates by genuinely teaching the model knowledge about the QuALITY corpus, rather than serving as a vehicle to distill a powerful prompted LM.

## 7.2 FUTURE DIRECTIONS

**Continued scaling beyond real data.** The large but finite body of human-written text is rapidly being consumed. Villalobos et al. (2024) predict that frontier language models will exhaust all public, human-generated text in 2028. As we transition from a data-rich to a data-constrained regime (Kaplan et al., 2020; Muennighoff et al., 2023), further scaling will require us to extract more knowledge from existing data. We demonstrated that synthetic continued pretraining with EntiGraph effectively extracts more knowledge from small corpora, which could help us learn from proprietary datasets or tail knowledge that appears only once or twice on the internet. It is an open question whether synthetic data generation methods like EntiGraph could improve data efficiency more generally on standard pretraining data and without relying upon a stronger prompted model.

**Alternatives to long-context language models.** Recent work handles long user queries (e.g., 1M-10M+ tokens) using efficient attention (Dao et al., 2022; Liu et al., 2023; Gemini, 2024) or architectures that are sub-quadratic in the context length (Tay et al., 2022; Gu et al., 2022; Gu & Dao, 2024; Sun et al., 2024). In settings where many queries share a long prefix—e.g., a corporation's proprietary documents or other prompt caching use cases (Anthropic, 2024a)—one could instead continue pretraining on the prefix to internalize its knowledge, and then perform standard quadratic attention on shorter queries. This approach pays a fixed training cost to amortize the prefix's knowledge into the weights of a model, and then benefits from shorter context lengths (Gururangan et al., 2020; Snell et al., 2022). By adapting the continued pretraining paradigm from 10B-100B tokens to as little as 1.3M tokens, our synthetic continued pretraining approach could enable unsupervised learning of shared text prefixes at much smaller and more practical token counts.

## 7.3 CONCLUSION

Continued pretraining with next-token prediction is remarkably effective in teaching pretrained language models new knowledge, but to date has only been applied successfully in broad, data-rich domains with 10B-100B+ tokens. We downscale continued pretraining to small, specialized corpora with ∼1M tokens using synthetic continued pretraining: converting a small corpus into a large synthetic one with diverse representations of knowledge, and continuing pretraining on it.

We instantiate this approach using EntiGraph, a knowledge graph–inspired synthetic data augmentation algorithm. Synthetic continued pretraining with EntiGraph demonstrates consistent scaling in downstream closed-book QA performance up to a 455M token synthetic corpus, whereas baselines such as continued pretraining on the small corpus or synthetic paraphrases show no improvement or scale slowly. Moreover, the acquired parametric knowledge composes with instruction tuning and retrieved non-parametric knowledge in an open-book setting. Lastly, we present a simplified mathematical model of EntiGraph and derive a functional form for its scaling trend, which closely matches our empirical trend. We hypothesize that EntiGraph's "externalization" of the synthetic data generation process to a combinatorial structure—in this case, a knowledge graph over entities—is a generally useful strategy in synthesizing highly diverse data and a promising object for future study.

## 8 ACKNOWLEDGEMENT

Zitong Yang would like to thank Samy Jelassi for feedback on a preliminary version of this work, Ruiqi Zhong for discussion regarding context distillation work, Xiang Lisa Li for discussion about reversal curse work, and the participants of the statistics seminar at Stanford University for their insightful feedback about a preliminary version of this work. We also thank the Tatsu Lab for constructive feedback and interesting discussions that have helped improve the paper. Zitong Yang is supported by the Albion Walter Hewlett Stanford Graduate Fellowship. Neil Band acknowledges funding from an NSF Graduate Research Fellowship and a Quad Fellowship. T.H. was supported by a grant from Samsung Research, gifts from Panasonic Research, the Google Research Scholar Program, and the Tianqiao and Chrissy Chen Institute, as well as the NSF grant IIS-2338866. E.J.C. is supported by the Office of Naval Research grant N00014-20-1-2157, the National Science Foundation grant DMS-2032014, the Simons Foundation under award 814641.

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

## CODEBASE, DATASET, AND MODEL WEIGHTS

We provide the codebase for reproducing all results discussed in the paper below:

```
https://github.com/ZitongYang/Synthetic_Continued_Pretraining.git
```

We release the 455M EntiGraph corpus below:

```
https://huggingface.co/datasets/zitongyang/entigraph-quality-corpus
```

We release the EntiGraph CPT model weights below:

```
https://huggingface.co/zitongyang/llama-3-8b-entigraph-quality
```

## A    ABLATION STUDIES

We present ablation experiments to further validate EntiGraph's effectiveness and test its generalization properties. We discussed two potential limitations in §7.1:

1. Could the gains of Synthetic CPT be explained by distillation effects, due to the use of a strong prompted LM for synthetic data generation?
2. Is the data synthesized in Synthetic CPT factual?

We provide evidence suggesting these are not significant concerns in Appendix A.1 and Appendix A.2, respectively. Lastly, we repeat the procedure of the core experiments on another small corpus of Coursera lecture transcripts, to provide evidence that Synthetic CPT generalizes to datasets and domains beyond QuALITY (Appendix A.3).

### A.1    USING A WEAKER SYNTHETIC DATA GENERATION LM

One potential concern is whether EntiGraph's success demonstrated in §4 stems from distilling knowledge from GPT-4. To investigate this, we conducted an experiment replacing GPT-4-Turbo with a significantly weaker model, Llama 3.1 8B Instruct, as the synthetic data generator. Recall that in all continued pretraining experiments, we finetune the 8B parameter Llama 3 Base model. Therefore, in this experiment, the capabilities of the synthetic data generator and the continually pretrained model are very similar, controlling for distillation effects. Using the entity extraction and relation analysis prompts introduced in §2, we generate 334M synthetic tokens and evaluate the scaling behavior under the same hyperparameter setup detailed in §4.1.

Figure 5 reveals two key insights. First, even with the weaker generator, EntiGraph maintains steady log-linear improvement with no signs of saturation at 334M tokens, suggesting that the gains of Synthetic CPT stem from continued pretraining on diverse representations of the corpora's underlying knowledge, rather than distilling the generator model's knowledge. Similar to our main results (§4), EntiGraph with a Llama 3.1 8B Instruct generator consistently outperforms Rephrase with the same generator. Moreover, at 334M synthetic tokens, EntiGraph with a Llama 3.1 8B Instruct generator outperforms closed-book evaluation of GPT-4-Turbo.

Second, while switching from the GPT-4-Turbo generator to the weaker generator shifts the accuracy curve downward, the log-linear slope remains consistent. In contrast, holding the synthetic generator constant, we observe that EntiGraph CPT and Rephrase CPT exhibit different slopes.

### A.2    FACTUALITY AND LEXICAL DIVERSITY OF ENTIGRAPH SYNTHETIC CORPUS

**Factuality.**    A limitation discussed in §7.1, and inherent in all methods involving synthetic data generation, is that the generation model may hallucinate. EntiGraph is a synthetic data *augmentation*, which conditions an LM on a given corpus document and prompts the LM to discuss the docu-

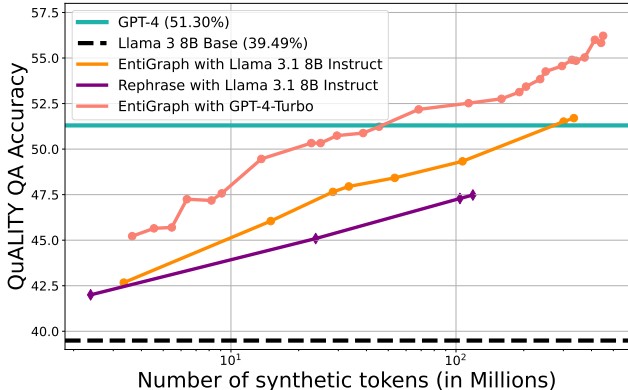

Figure 5: The scaling properties of Synthetic CPT with the EntiGraph and Rephrase augmentations, comparing two synthetic data generators: GPT-4-Turbo and Llama 3.1 8B Instruct.

ment's entities and their relationships. Assuming a reasonably good generator model, this grounding should decrease hallucination rate.

To quantitatively test the factuality of documents synthesized with EntiGraph, we split the 455M token EntiGraph corpus into sentences and randomly sample 150 sentences. We ask authors of this work to label whether each sentence is subjective or not, and among non-subjective sentences, to determine whether it is supported by the article text or not.

We compute two statistics: the proportion of subjective sentences denotes the number of subjective sentences over the total number of annotated sentences. The factuality rate denotes the number of non-subjective sentences which are supported by the source document, over the number of non-subjective sentences, following Min et al. (2023):

- Proportion subjective: $0.532$ (bootstrap $0.95$ confidence interval: $[0.455, 0.610]$).

- Factuality rate: $0.944$ (bootstrap $0.95$ confidence interval: $[0.889, 0.986]$).

Because EntiGraph uses open-ended prompts which ask the LM to relate different, often abstract entities, the LM often generates subjective statements. We do not necessarily view this as a limitation, because learning reasonable subjective interpretations is crucial for understanding (and hence is often assessed in, e.g., essay questions on literature exams). We also observe that the non-subjective sentences are consistently factual, supporting the effectiveness of grounding in reducing hallucination.

**Lexical Diversity.** We hypothesize that good synthetic data augmentations should produce knowledge representations with diverse wording. As a measure of this lexical diversity, we compute the percentage of $n$-grams in the synthetic documents that overlap with the $n$-grams of the corresponding source documents.

More precisely, we first randomly select 100 QuALITY articles, tokenize them with the Llama 3.1 tokenizer, and compute the set of $n$-grams for each article. Then, for each article, we tokenize the corresponding EntiGraph and Rephrase synthetic data, compute $n$-grams, and count the $n$-grams in the synthetic data that appear in the set of $n$-grams for the raw article. For each $n$ and synthetic augmentation method, we sum this overlap count across articles and normalize by the total number of synthetic tokens generated for the 100 articles, providing us an estimate of the percentage of $n$-grams in the synthetic data that overlap with the source data.

These results are provided in Table 4. We observe that for both augmentations, $n$-gram overlap percentage is low and quickly approaches $0\%$ with increasing $n$, indicating that both methods produce lexically diverse knowledge representations.

| Augmentation | $n = 2$ | $n = 4$ | $n = 8$ | $n = 16$ |
|---|---|---|---|---|
| EntiGraph | 23.40 | 3.66 | 0.24 | 0.00 |
| Rephrase | 21.35 | 3.04 | 0.51 | 0.22 |

Table 4: Percentage of token $n$-grams in synthetic documents that overlap with the source document $n$-grams, for the EntiGraph and Rephrase synthetic data augmentations.

### A.3 DATASETS BEYOND QuALITY

To test whether synthetic CPT with EntiGraph generalizes to corpora beyond QuALITY, we evaluated on the Coursera Exam QA dataset (An et al., 2023). This dataset contains lecture transcripts and exam questions from advanced technical courses like data science and machine learning. Compared to the books and stories in QuALITY, Coursera exams present new challenges—the content is harder conceptually, questions can have multiple correct answers, and the number of options is not fixed to four choices. This makes few-shot prompting more demanding, as the model must understand both the content and the flexible answering format.

The dataset consists of 15 lecture transcripts and 124K raw tokens, substantially smaller than QuALITY's 265 documents and 1.3M raw tokens. During our scaling analysis, we found that models trained on tiny synthetic corpora (e.g., a few million tokens) struggled to follow few-shot prompts reliably for Coursera questions, resulting in parsing errors. Therefore, we begin the scaling curve in Fig. 6 starting from token counts where parsing error rates fall below 5%. For the Rephrase baseline, we generate synthetic data up to 22M tokens, and find that only one model has parsing error rates below 5%.

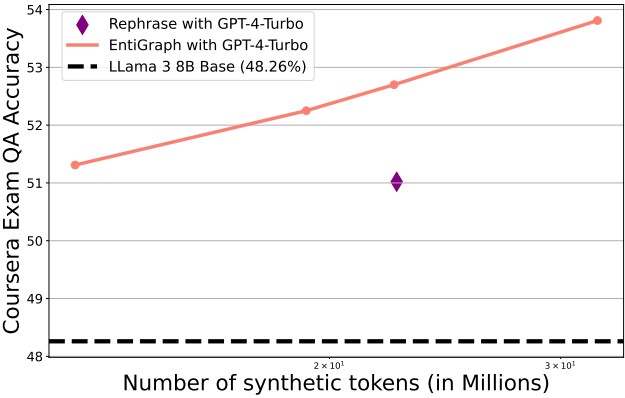

Figure 6: The scaling properties of Synthetic CPT using the EntiGraph augmentation on the Coursera Exam QA dataset.

Despite these challenges, EntiGraph CPT shows consistent improvement over Llama 3 8B Base, improving accuracy from 48.26% to 53.87%, better than Llama 3 8B Base and the Rephrase baseline. The log-linear scaling pattern persists up to 32M synthetic tokens, suggesting EntiGraph's effectiveness extends beyond narrative texts to technical educational content. This successful transfer to a substantially different domain provides evidence for the generalizability of synthetic continued pretraining and EntiGraph.

## B  ADDITIONAL RELATED WORK

**Synthetic data generation.**    There is a rich literature on using neural nets to generate synthetic data. Many such approaches were originally developed for semi-supervised learning—self-training and pseudo-labeling methods improve models by iteratively training them on their own predictions (Scudder, 1965; Lee, 2013; Yalniz et al., 2019; Berthelot et al., 2019; Xie et al., 2020), and co-training uses two models to supervise each other (Blum & Mitchell, 1998; Balcan et al., 2004). Before language models rose to prominence, few approaches attempted to synthesize inputs. One exception is membership query synthesis, which explored the synthesis of inputs in a supervised learning context (Angluin, 1988; Schumann & Rehbein, 2019).

Contemporary works employ co-training (Lang et al., 2022) and self-training to improve language model performance, often on mathematical reasoning tasks (Huang et al., 2023; Gulcehre et al., 2023; Zhang et al., 2024a), or synthesize input-output pairs for instruction tuning, usually by conditioning on a curated seed set (Wang et al., 2023b; Honovich et al., 2023; Taori et al., 2023; Peng et al., 2023; Yuan et al., 2024b; Li et al., 2024).

**Continual learning and pretraining.**    Continual learning is rooted in historical work on connectionist networks (McCloskey & Cohen, 1989; Ratcliff, 1990) and considers learning with tasks arriving in an online manner (Schlimmer & Fisher, 1986; Grossberg, 2012). The main focus is on mitigating a neural net's "catastrophic forgetting" of previously encountered tasks (Robins, 1995; Goodfellow et al., 2015; Kemker et al., 2018). Approaches include regularizing parameter updates to preserve important parameters (Nguyen et al., 2017; Zenke et al., 2017; Kirkpatrick et al., 2017); dynamically modifying the architecture (Rusu et al., 2016; Golkar et al., 2019); and recalling or replaying previous experiences (Rebuffi et al., 2017; Shin et al., 2017; Lopez-Paz & Ranzato, 2017). Modern works in continued pretraining (cf. §1.1) effectively mitigate catastrophic forgetting by scaling parameter count (Ramasesh et al., 2022) and mixing in updates on pretraining data (Ouyang et al., 2022).

## C  DETAILS ON THE QuALITY DATASET

We provide additional details on the QuALITY dataset below. For each book, we execute entity extraction (Step 1, §2.2) and then analyze all pair-wise relations between entities and a subset of all triplet relations (Step 2, 2.2). We provide summary statistics for the Raw and EntiGraph corpora in Figure 7.

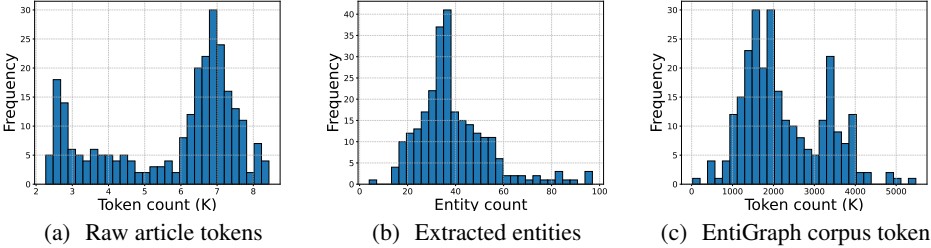

(a)  Raw article tokens          (b)  Extracted entities          (c)  EntiGraph corpus tokens

Figure 7: Histograms over the 265 QuALITY articles and books. (a) The token count of raw articles. (b) The number of extracted entities. (c) The token count of EntiGraph synthetic data (generated for each book).

## D  TRAINING DETAILS FOR THE MAIN EXPERIMENTS

**Continued pretraining details.**    In all experiments, we continue pretraining the Llama 3 8B Base model with a context length of 2048 and batch size of 16. We apply a linear learning rate warmup for 5% of total steps, followed by a cosine decay with peak learning rate 5e-6. We perform full parameter training with Fully Sharded Data Parallelism (FSDP, Zhao et al. (2023)).

**EntiGraph continued pretraining details.**  To mitigate the forgetting of pretrained knowledge, we perform replay with a rate of 0.1 using 1B RedPajama tokens (TogetherAI, 2023). More precisely, for each training batch, we flip a biased coin such that with 10% probability, we load the RedPajama data instead of the EntiGraph synthetic data.

**Raw continued pretraining details.**  Next, we provide details for our continued pretraining directly on the Raw corpus, producing the "Raw CPT" model. Because the Raw corpus only has 1.3M tokens, we jointly tune the number of epochs (repetition factor) and the RedPajama replay rate on accuracy over a QuALITY QA validation split. The selected hyperparameter configuration uses 4 epochs and a 0.1 replay rate.

**Instruction tuning details.**  We use the UltraChat instruction tuning dataset (Ding et al., 2023) filtered by the Huggingface team (Tunstall et al., 2023) as our instruction tuning data. We use the chat template of Llama 3.1 8B Instruct (Dubey et al., 2024) to format the UltraChat conversations, obtaining a 250M token instruction tuning dataset. We apply a linear learning rate warmup followed by a cosine decay to 0 with peak learning rate 5e-6, and train the model for 1 epoch with a batch size of 512 and context window of 2048. To sanity check our instruction tuning procedure, we measure the AlpacaEval (Li et al., 2023a) winrate against GPT-4 and find it improves from 0% to 6.25%, comparable to a 7.7% baseline winrate of Llama 2 Chat 13B.

**Compute resource.**  All the continued pretraining experiments are performed with one $8 \times$H100 node. With PyTorch FSDP (Zhao et al., 2023), we obtain throughput of 6090 tokens per second. Since all experiments use the same model architecture, batch size, and context length, the time to run the experiments can be calculated based on the total tokens seen during training. For example, the main EntiGraph is trained on 455M tokens with 2 epochs. Therefore, it should take $455M \times 2/6090$ seconds, which is about 41 hours.

## E    TASK-SPECIFIC FINETUNING FOR THE QuALITY QUESTION SET

Our work considers *task-agnostic* synthetic data generation and continued pretraining as a way to obtain generalizable knowledge about a domain, in a way that can later be extracted via few-shot prompting (Brown et al., 2020) and instruction tuning (Ouyang et al., 2022).

However, if our goal is only to do well on a single task, such as question answering, then we could fine-tune a language model for that particular task. This approach worked extremely well on tasks such as SQuAD (Rajpurkar et al., 2016) in-domain but suffered from degraded performance outside the fine-tuning data distribution (Awadalla et al., 2022).

We do not extensively perform comparisons to task-specific finetuning due to the more general multi-task goals of EntiGraph. We run preliminary experiments comparing a simple QA SFT baseline to EntiGraph, and find that EntiGraph scaling and synthetic data generation costs are generally favorable even when compared to this strong, task-specific baseline.

**QA SFT.**  We follow the same set as in §2.1 and §3 except that we do not prompt $\mathsf{LM}_{\mathrm{synth}}$ to generate general knowledge about QuALTY articles. Instead, we prompt $\mathsf{LM}_{\mathrm{synth}}$ to generate QA pairs directly:

```
You are an assistant to help read a article and then rephrase it in a
question answering format. The user will provide you with an article
with title, year, content. You need to generate a paraphrase of the
same article in question and answer format with multiple tags of
"Question: ..." followed by "Answer: ...". Remember to keep the
meaning and every content of the article intact, including the title,
year, etc.
```

We repeat this prompt many times at temperature 1.0, resulting in 28M tokens on synthetic question answer pairs. We perform the same continued pretraining procedure in §4.1 on Llama 3 8B and refer to this model as "QA SFT".

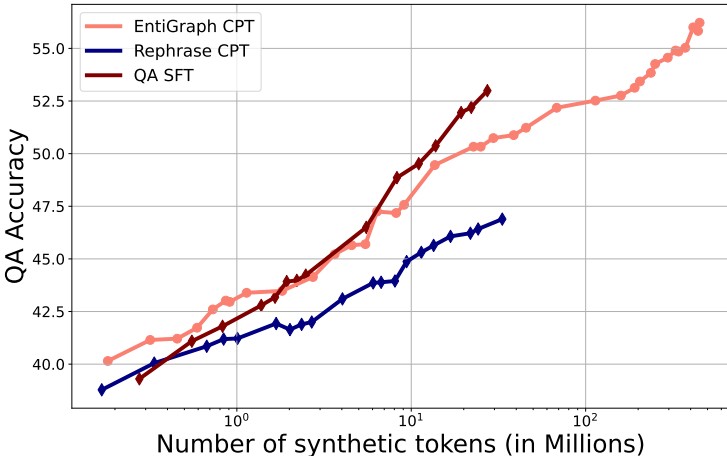

Figure 8: Accuracy on the QuALITY question set $\mathcal{Q}_{\text{test}}$ ($y$-axis) as a function of the synthetic token count ($x$-axis). Comparison among EntiGraph CPT, Rephrase CPT, and QA SFT.

**Results discussion**  We plot the QA SFT scaling curve in Figure 8. We can see that task-specific finetuning demonstrates a very sharp improvement in QA accuracy, consistent with prior results showing task-specific finetuning gains for pretrained models. While QA SFT performance is high, we note that EntiGraph attains similar performance despite being entirely task-agnostic, and the overall dollar cost of creating the dataset is much lower for EntiGraph.

This difference in synthetic data generation cost is hidden in Figure 8, as we plot the number of training tokens rather than dollars spent to generate the synthetic data. For QA SFT, each QA question is generally short, resulting in large inefficiencies in generating this QA dataset. We found that the input token to output token ratio was large compared with Rephrase CPT and EntiGraph CPT, resulting in over \$5K to generate just 28M tokens[4]. This difference in cost means that further scaling became prohibitively expensive, and that EntiGraph's performance in Figure 8 is even better than it appears, if we match for total cost rather than token budget.

## F  ADDITIONAL DETAILS ON OPEN-BOOK EXPERIMENTS

We provide additional details on our open-book experimental setup below, including our retrieval-augmented generation (RAG, Lewis et al. (2020); Gao et al. (2024)) pipeline. As mentioned in §5, we use a standard two-stage RAG pipeline: first, an offline stage which indexes document chunks; second, inference-time retrieval, reranking, and placement of those chunks in a few-shot LM prompt.

### F.1  STAGE 1: OFFLINE INDEXING

The purpose of the indexing stage is to construct an index over all the 265 articles and books from the QuALITY corpus $\mathcal{D}_{\text{source}}$. More specifically, this stage chunks documents from the given corpus, obtains dense vector embeddings for each chunk using an API-based embedding model, and indexes the (embedding, chunk) pairs.

**Chunking documents.**  We first split each document $D^{(i)} \in \{D^{(i)}\}_{i=1}^{n} = \mathcal{D}_{\text{source}}$ into a set of $m_i$ document chunks $\{C_1^{(i)}, ..., C_{m_i}^{(i)}\}$. To perform this splitting, we use the `Recursive CharacterTextSplitter` from Chase (2022), which attempts to keep all paragraphs (and then sentences, and then words) together for as long as possible, in order to preserve the semantics within each chunk. We use non-overlapping chunks and tune chunk size in characters (`chunk_size`, hyperparameter values provided below). Lastly, because we have access to metadata about each document $D^{(i)}$—namely, the title, author, and year of the book or article—we prepend this meta-data to each document chunk. This is analogous to how a corporation building a RAG system over

---

[4]OpenAI API pricing, Sep 2024

their own document store could include metadata about the document (title, author, year, etc.). These final chunks with metadata prepended are embedded, and are the ones that are retrieved and placed in-context.

**Embedding and indexing document chunks.** Next, we obtain dense embeddings for all document chunks using a state-of-the-art text embedding model OpenAI `text-embedding -3-large` (Neelakantan et al., 2022). Lastly, we index all (embedding, chunk) tuples using a FAISS vector store (Douze et al., 2024).

### F.2 STAGE 2: INFERENCE-TIME RETRIEVAL AND RERANKING

At inference time, the RAG system receives a test query $q \in \mathcal{Q}_{\text{test}}$. Each query $q$ is contextualized with the article title and author name, as described in §3, and contains its four possible answer choices (QuALITY is a 4-choice, multiple choice dataset). In Stage 2, we embed the query with the API-based embedding model, retrieve $K$ document chunks using an approximate nearest-neighbor search, and lastly, select the $k < K$ most relevant chunks using an API-based reranker.

**Retrieving top-$K$ document chunks.** We embed $q$ with `text-embedding-3-large`, and retrieve the top-$K$ most relevant document chunks from our indexed vector store using FAISS similarity search with a Euclidean distance metric.

**Reranking to obtain top-$k$ ($k < K$) chunks.** Next, we use a reranker to filter the $K$ retrieved document chunks to a smaller number of reranked chunks $k$. Rerankers are known to significantly improve recall (the proportion of the time that the salient article is contained in the top chunks), and indeed, the recall of our RAG pipelines is near-perfect (Table 3 in §5). Specifically, we pass the query $q$ and the list of $K$ retrieved document chunks to a state-of-the-art reranker—Cohere `rerank-english-v3.0` (Cohere, 2024)—which returns a list of the $K$ chunks in order from most to least semantically relevant for the query. We take the $k$ highest scoring chunks and place them in our few-shot prompt.

**Few-shot prompt formatting.** Our full few-shot chain-of-thought evaluation prompts for the open-book setting will be provided in our code release. Similar to the closed-book QA evaluation prompt, we manually write and fact-check in-context learning examples about well-known books, to avoid leaking knowledge from the QuALITY articles. In early experiments, we found that placing the retrieved contexts first, followed by the question and answer choices after, significantly improved performance compared to question-then-contexts; we use this format throughout the retrieval experiments. We treat as a hyperparameter whether the reranked chunks are ordered from the best match to worst (`best_first`) or from the worst match to best (`best_last`). When performing few-shot evaluation, we follow the sampling procedure used in the closed-book experiments (Appendix I.1). Specifically, we generate 64 responses for each question, and filter out responses that do not parse to one of the four choices. Lastly, we randomly select one of the valid responses as the model's final answer.

### F.3 HYPERPARAMETER TUNING

In our experiments, we compare two LMs used in the RAG pipeline above: EntiGraph CPT and its base model, Llama 3 8B Base. As mentioned above, we fix the retrieved number of chunks to $K = 128$, but vary the number of reranked chunks $k$ which are ultimately placed in the context window. For each language model + RAG pipeline, we independently tune the following hyperparameters with a grid search on accuracy using a QuALITY QA validation split:

- Document `chunk_size` $\in \{256, 512, 1024\}$
- Rerank top-$k \in \{1, 2, 4, 8, 16\}$
- Order of chunks $\in \{$`best_first`, `best_last`$\}$
- Eval temperature $\in \{0.1, 0.3, 0.5, 0.7\}$

We will provide tuned hyperparameters in our code release.

## G  PROOF OF THEOREM 1 AND OTHER ANALYTICAL FORMULAS

In this section, we prove Theorem 1 and provide the derivations for several other approximation formulas.

*Proof of Theorem 1.* Fix the matrix $\boldsymbol{M}_0$, we observe that

$$\mathsf{Acc}(\boldsymbol{M}_t) = \frac{\mathbb{E}[\|\boldsymbol{M}_t\|_1|\boldsymbol{M}_0]}{V(V-1)} = \sum_{(i,j)\in\mathcal{V}^2} \frac{\mathbb{E}[\mathbb{1}((i,j)\in\mathcal{D}_t)|\boldsymbol{M}_0]}{V(V-1)} = \sum_{(i,j)\in\mathcal{V}^2} \frac{\mathbb{P}[(i,j)\in\mathcal{D}_t|\boldsymbol{M}_0]}{V(V-1)}.$$

For each $(i,j) \in \mathcal{V}^2$, we define $q_{i,j}$ to be the probability that $(i,j)$ is included in the set $\{(x_t, z_t^1), (x_t, z_t^2), \ldots, (x_t, z_t^{k_t}), (x_t, y_t)\}$. Note that each iteration of the procedure generates a path $(x_t, z_t^1, z_t^2, \ldots, z_t^{k_t}, y_t)$ independently identically. So naturally $q_{i,j}$ does not depend on the time $t$. This implies that $\mathbb{P}[(i,j) \in \mathcal{D}_t|\boldsymbol{M}_0] = 1 - (1 - q_{i,j})^t$. Thus we can further rewrite the link density as

$$\mathsf{Acc}(\boldsymbol{M}_t) = \frac{|\mathcal{D}_{\text{source}}|}{V(V-1)} + \sum_{(i,j)\in\mathcal{V}^2\setminus\mathcal{D}_{\text{source}}} \frac{\mathbb{P}[(i,j)\in\mathcal{D}_t|\boldsymbol{M}_0]}{V(V-1)}$$

$$= \frac{|\mathcal{D}_{\text{source}}|}{V(V-1)} + \sum_{(i,j)\in\mathcal{V}^2\setminus\mathcal{D}_{\text{source}}} \frac{1-(1-q_{i,j})^t}{V(V-1)}.$$

The remaining task is to estimate $q_{i,j}$. We say a vertex $j$ is reachable from $i$ and denote $i \sim j$, if there is a directed path from $i$ to $j$ in $\boldsymbol{M}_0$. We define $\mathcal{R} = \{(u,v) \in \mathcal{V}^2 : u \neq v, u \sim v\}$ to be the set of all reachable pairs of vertices in $\mathcal{V}$. We note that $q_{i,j}$ is non-zero if and only if $j$ is reachable from $i$ in $\boldsymbol{M}_0$. Now, for any $t \geq 1$, the function $1 - (1-x)^t$ is concave, thus by Jensen's inequality, we have

$$\sum_{(i,j)\in\mathcal{V}^2\setminus\mathcal{D}_{\text{source}}} 1 - (1-q_{i,j})^t \leq \sum_{(i,j)\in\mathcal{R}} 1 - (1-q_{i,j})^t \leq |\mathcal{R}| \left(1 - (1-\bar{q}_{i,j})^t\right),$$

where

$$\bar{q}_{i,j} = \frac{\sum_{(i,j)\in\mathcal{R}} q_{i,j}}{|\mathcal{R}|}.$$

For each $(i,j) \in \mathcal{R}$, the probability $q_{i,j}$ satisfies

$$q_{i,j} = \frac{\sum_{a\neq b\in\mathcal{V}^2} \mathbb{1}((i,j) \in \{(a,z^1),(a,z^2),\ldots,(a,z^k),(a,b)\})}{V(V-1)}$$

where $(a, z^1, z^1, \cdots, z^k, b)$ is the shortest path in $\boldsymbol{M}_0$ connecting $a$ and $b$. If there is no such path, then by default the indicator equals zero. Now we look at

$$\sum_{(i,j)\in\mathcal{R}} q_{i,j} = \frac{1}{V(V-1)} \sum_{(i,j)\in\mathcal{R}} \sum_{(a,b)\in\mathcal{R}} \mathbb{1}((i,j) \in \{(a,z^1),(a,z^2),\ldots,(a,z^k),(a,b)\})$$

$$\leq \frac{1}{V(V-1)} \sum_{(a,b)\in\mathcal{R}} \sum_{i\neq j\in\mathcal{V}^2} \mathbb{1}((i,j) \in \{(a,z^1),(a,z^2),\ldots,(a,z^k),(a,b)\})$$

$$= \frac{1}{V(V-1)} \sum_{(a,b)\in\mathcal{R}} \ell_{a,b},$$

where $\ell_{a,b}$ is the length of the shortest path connecting $a$ to $b$. To analyze the typical shortest length of paths, we present a few classical results on directed Erdős-Rényi graphs. For any $a \in \mathcal{V}$, let $X(a)$ denote the set of vertices reachable from $a$ and let $Y(a)$ denote the set of vertices from which $a$ is reachable. Recall that $\rho(\lambda)$ is the extinction probability for the Poisson$(\lambda)$ branching process.

**Lemma G.1** (Lemma 1 and Corollary 1 in Karp (1990)). *For each vertex $a$, with probability tending to 1 as $V$ tends to infinity, there exists a constant $\beta > 0$ such that either $|X(a)| \leq \beta \log V$ or $|X(a)| = (1-\rho(\lambda))V + \Theta(\sqrt{V})$. Moreover, the probability that the latter happens tends to $1-\rho(\lambda)$ as $V$ tends to infinity. The same is true for $Y(a)$.*

For each vertex $a$, the set $X(a)$ is said to be small if $|X(a)| \leq \beta \log V$ (in such case we write $a \in \mathcal{S}_X$) and large if $|X(a)| = (1 - \rho(\lambda))V + \Theta(\sqrt{V})$ (we write $a \in \mathcal{L}_X$). We define $\mathcal{S}_Y$ and $\mathcal{L}_Y$ similarly.

**Lemma G.2** (Theorem 3 in Karp (1990) and Theorem 2.4.1 in Durrett (2010)). *With probability tending to* 1*, the following statement holds for all $a$ and $b$ in $\mathcal{V}$: if $X(a)$ is large and $Y(b)$ is large, then $b$ is reachable from $a$. Moreover, if $X(a)$ is large and $Y(b)$ is large, then for any $\varepsilon > 0$ and any sufficiently small $\delta > 0$,*

$$\mathbb{P}[\ell_{a,b} > (1 + \varepsilon) \log V / \log \lambda] < \exp(-V^\varepsilon \delta).$$

With Lemma G.1 and Lemma G.2, we can now give useful estimates of $|\mathcal{R}|$. In particular, for any $\varepsilon > 0$,

$$\begin{aligned}
|\mathcal{R}| &= |\{(a, b) \in \mathcal{R} : a \in \mathcal{L}_X, b \in \mathcal{L}_Y\}| + |\{(a, b) \in \mathcal{R} : a \in \mathcal{S}_X \text{ or } b \in \mathcal{S}_Y\}| \\
&\leq (1 - \rho(\lambda))^2 (1 + \varepsilon/4) V^2 + 2(1 + \varepsilon) V \beta \log V \\
&\leq (1 - \rho(\lambda))^2 (1 + \varepsilon/3) V(V - 1),
\end{aligned}$$

with high probability. Similarly, for the lower bound,

$$\begin{aligned}
|\mathcal{R}| &= |\{(a, b) \in \mathcal{R} : a \in \mathcal{L}_X, b \in \mathcal{L}_Y\}| + |\{(a, b) \in \mathcal{R} : a \in \mathcal{S}_X \text{ or } b \in \mathcal{S}_Y\}| \\
&\geq (1 - \rho(\lambda))^2 (1 - \varepsilon) V^2 \\
&\geq (1 - \rho(\lambda))^2 (1 - \varepsilon) V(V - 1),
\end{aligned}$$

with high probability. By a union bound over all pairs of $(a, b) \in \mathcal{R}$, we also have that

$$\begin{aligned}
\sum_{(i,j) \in \mathcal{R}} q_{i,j} &\leq \frac{1}{V(V - 1)} \sum_{(a,b) \in \mathcal{R}} \ell_{a,b} \\
&= \frac{1}{V(V - 1)} \sum_{\substack{(a,b) \in \mathcal{R} \\ a \in \mathcal{L}_X, b \in \mathcal{L}_Y}} \ell_{a,b} + \frac{1}{V(V - 1)} \sum_{\substack{(a,b) \in \mathcal{R} \\ a \in \mathcal{S}_X \text{ or } b \in \mathcal{S}_Y}} \ell_{a,b} \\
&\leq (1 - \rho(\lambda))^2 (1 + \varepsilon/2) \frac{\log V}{\log \lambda} + \frac{1}{V(V - 1)} 2(1 + \varepsilon) V (\beta \log V)^2 \\
&\leq (1 - \rho(\lambda))^2 (1 + \varepsilon) \frac{\log V}{\log \lambda},
\end{aligned}$$

with probability larger than $1 - V^2 \exp(-V^\varepsilon \delta)$. Combining the above, for any $\varepsilon > 0$,

$$\bar{q}_{i,j} = \frac{\sum_{(i,j) \in \mathcal{R}} q_{i,j}}{|\mathcal{R}|} \leq \frac{(1 + \varepsilon) \log V}{V(V - 1) \log \lambda},$$

with high probability. Therefore, for any $\varepsilon > 0$,

$$\begin{aligned}
\mathsf{Acc}(\boldsymbol{M}_t) &\leq \frac{|\mathcal{D}_{\text{source}}|}{V(V - 1)} + \frac{|\mathcal{R}| (1 - (1 - \bar{q}_{i,j})^t)}{V(V - 1)} \\
&\leq (1 + \varepsilon) \left( p + (1 - \rho(\lambda))^2 \left( 1 - \left( 1 - \frac{(1 + \varepsilon) \log V}{V(V - 1) \log \lambda} \right)^t \right) \right),
\end{aligned}$$

with high probability, which completes the proof of the upper bound. For the lower bound, we observe that if $i \sim j$ and $(i, j) \in \mathcal{R} \backslash \mathcal{D}_{\text{source}}$, then $q_{i,j} \geq 1/V(V - 1)$, because when $i$ and $j$ are chosen in the procedure, the edge $(i, j)$ will be added. This implies that

$$\begin{aligned}
\mathsf{Acc}(\boldsymbol{M}_t) &= \frac{|\mathcal{D}_{\text{source}}|}{V(V - 1)} + \sum_{\mathcal{R} \backslash \mathcal{D}_{\text{source}}} \frac{1 - (1 - q_{i,j})^t}{V(V - 1)} \\
&\geq \frac{|\mathcal{D}_{\text{source}}|}{V(V - 1)} + \frac{|\mathcal{R} \backslash \mathcal{D}_{\text{source}}|}{V(V - 1)} \left( 1 - \left( 1 - \frac{1}{V(V - 1)} \right)^t \right) \\
&\geq (1 - \varepsilon) \left( p + (1 - \rho(\lambda))^2 \left( 1 - \left( 1 - \frac{1}{V(V - 1)} \right)^t \right) \right),
\end{aligned}$$

with high probability which completes the proof of the lower bound. $\qquad\square$

To obtain a more precise description of $\mathsf{Acc}(\boldsymbol{M}_t)$, we employ a Poisson branching process to approximate the cluster growth of vertices, which we now define. A Poisson($\lambda$) branching process is a model for a population evolving in time, where each individual independently gives birth to a number of children with Poisson($\lambda$) distribution. We denote by $Z_n$ the number of individuals in the $n$-th generation, where by default $Z_0 = 1$. Then $Z_n$ satisfies the recursion relation $Z_n = \sum_{i=1}^{Z_{n-1}} X_{n,i}$, where $\{X_{n,i}\}_{n,i \geq 1}$ is a doubly infinite array of i.i.d. Poisson($\lambda$) random variables. The total progeny $Y_n$ is then defined as $Y_n = \sum_{i=0}^{n} Z_n$. $Z_n$ is often called a Galton–Watson branching process and the associated tree is called a Galton–Watson tree.

As in the previous proof, an accurate estimate of $\mathsf{Acc}(\boldsymbol{M}_t)$ relies on understanding $q_{i,j}$, the probability that the edge $(i, j)$ will be added in each round. As before, the only edges that will be added are those connected to the giant component (i.e., $i \in \mathcal{L}_X$ and $j \in \mathcal{L}_Y$). The proportion of such edges converges to $C_\lambda$ as $V \to \infty$. Recall that

$$q_{i,j} = \frac{\sum_{(a,b) \in \mathcal{R}} \mathbb{1}((i,j) \in \{(a, z^1), (a, z^2), \ldots, (a, z^k), (a, b)\})}{V(V-1)} \tag{3}$$

where $(a, z^1, z^1, \cdots, z^k, b)$ represents the shortest path in $\boldsymbol{M}_0$ connecting $a$ and $b$. Equivalently, if we consider the tree generated by a breadth-first search in $\boldsymbol{M}_0$ rooted at $i$, then since $i \sim j$, $j$ will be in the tree, and the numerator counts the total number of offspring of $j$ in the tree, including $j$ itself. This is the point at which a rigorous mathematical characterization of the tree becomes challenging. Instead, we approximate the tree and analyze its behavior. It is well-known that when $p = \lambda/V$, the cluster growth (or the breadth-first search at a vertex) can be approximated by a Poisson($\lambda$) branching process (see e.g., Hofstad (2016); Durrett (2010)). For fixed vertex $i$, we define $T$ as a Galton–Watson tree rooted at $i$ with Poisson($\lambda$) offspring distribution with depth $L$. We use $T$ to approximate the exploration process at $i$. For $0 \leq \ell \leq L$, the number of vertices at level $L - \ell$ is approximately $\lambda^{L-\ell}$. Given that the total number of vertices in $T$ is approximately $(1 - \rho(\lambda))V$, the number of vertices at level $L - \ell$ is also $(1 - \rho(\lambda))V(\lambda - 1)/\lambda^{\ell+1}$. For each vertex at level $L - \ell$, the number of its offspring (including itself) equals $k$ with probability $p_\ell(k)$. In this case, the numerator in (3) equals $k$. Combining the above, there are around $(1 - \rho(\lambda))V \cdot p_\ell(k)(1 - \rho(\lambda))V(\lambda - 1)/\lambda^{\ell+1}$ vertex pairs $(i, j)$ in the graph such that $i \in \mathcal{L}_X$, $j \in \mathcal{L}_Y$, $q_{i,j} = k/V(V-1)$ and $j$ is located at the $L - \ell$ level in the tree $T$. Ultimately, we arrive at an approximation of the form

$$\mathsf{Acc}(\boldsymbol{M}_t) \sim p + C_\lambda \left( 1 - \sum_{\ell=0}^{\infty} \frac{\lambda - 1}{\lambda^{\ell+1}} \sum_{k=1}^{\infty} p_\ell(k) \left( 1 - \frac{k}{V(V-1)} \right)^t \right).$$

Beyond Erdős-Rényi graphs, the term $q_{i,j}$ may not be as explicit. We can define $C$ as the proportion of vertex pairs $(i, j)$ such that $i \sim j$ in $\boldsymbol{M}_0$, then $q_{i,j}$ is nonzero for $CV(V-1)$ pairs of vertices. In this case, if we write $a_k = k/V(V-1)$ and define $\mu(k)$ as the probability that $q_{i,j} = a_k$, then we can have a general formula

$$\mathsf{Acc}(\boldsymbol{M}_t) \sim p + C \left( 1 - \sum_{k=1}^{\infty} \mu(k) (1 - a_k)^t \right).$$

The drawback of this formula is the lack of explicit expressions. For a given $\boldsymbol{M}_0$, it is unclear how to compute the measure $\mu(\cdot)$ easily.

Next, we provide a qualitative description of the shape of such a mixture of exponentials.

**Lemma G.3.** *For a fixed constant $0 < C < 1$ and a probability measure $\mu(\cdot)$ on $\mathbb{Z}_+$ with finite mean $m$, we define*

$$f(t) = p + C \left( 1 - \sum_{k=1}^{\infty} \mu(k) \left( 1 - \frac{k}{V(V-1)} \right)^{tV(V-1)} \right).$$

*Then we have that there exists $0 < t_1 < t_2$ such that*

$$f(t) = \begin{cases} \Theta(p + t), & \text{for } 0 \leq t \leq t_1, \\ \Theta(\log t), & \text{for } t_1 \leq t \leq t_2, \\ \Theta(1), & \text{for } t \geq t_2, \end{cases}$$

*as $V \to \infty$.*

*Proof of Lemma G.3.* Fix any $1 < t_1 < t_2$. Note that $f(t)$ is monotone increasing, concave and always bounded by 1. We also have

$$f(t_2) \geq p + C\left(1 - \left(1 - \frac{1}{V(V-1)}\right)^{t_2 V(V-1)}\right) \geq p + C(1 - \exp(-t_2)) = \Theta(1).$$

So $f(t) = \Theta(1)$ when $t \geq t_2$. Now when $t \leq t_1$,

$$f(t) \leq p + C\left(1 - \sum_{k=1}^{\infty} \mu(k)(1 - tk)\right) \leq p + Cmt.$$

Since $f(0) = p$ and $f(t_2) \geq p + C(1 - \exp(-t_2))$, by concavity, $f(t)$ is lower bounded by $p + tC(1 - \exp(-t_2))/t_2 = \Theta(p + t)$ for any $0 \leq t \leq t_1$. Finally for $t_1 \leq t \leq t_2$, we note that $f(t_1) \leq f(t) \leq 1$, so easily, $f(t) \leq \log t_1 / \log t_1 \leq \log t / \log t_1 = O(\log t)$. Similarly, $f(t) \geq f(t_1) \log t_2 / \log t_2 \geq \log t (f(t_1)/\log t_2) \geq \Omega(\log t)$. Therefore, $f(t) = \Theta(\log t)$ for any $t_1 \leq t \leq t_2$. $\square$

### G.1 MORE DETAILS ON THE MIXTURE OF EXPONENTIAL SHAPE

We provide more discussion on the mixture of exponential shape, including how we use it to fit the empirical EntiGraph CPT QA accuracy.

**Sketch of derivation.** Intuitively, the edge $(i, j)$ will eventually be added if and only if $j$ is reachable from $i$ in the original graph $M_0$. This explains the limiting behavior of $\text{Acc}(M_t)$ as $t$ approaches infinity: the proportion of links will converge to the proportion of connected vertex pairs in $M_0$. To understand the mixture-of-exponential functional form, consider that at the time $t$, the probability of adding each vertex pair follows an exponential pattern, with different vertex pairs exhibiting different exponential growth rates. Specifically, think of a breadth-first search in $M_0$ starting from a vertex $i$. If $j$ is very close to the root, there are many paths from $i$ to other vertices passing through $j$, making it more likely that $(i, j)$ will be included in each iteration. In contrast, if $j$ is far from the root (e.g., at the end of the exploration process), there are fewer such paths, making it less likely for $(i, j)$ to be included in each iteration. This accounts for the mixture-of-exponential shape, where the mixture primarily reflects the distance of each vertex from the root, the number of such vertices, and their corresponding exponential growth rates.

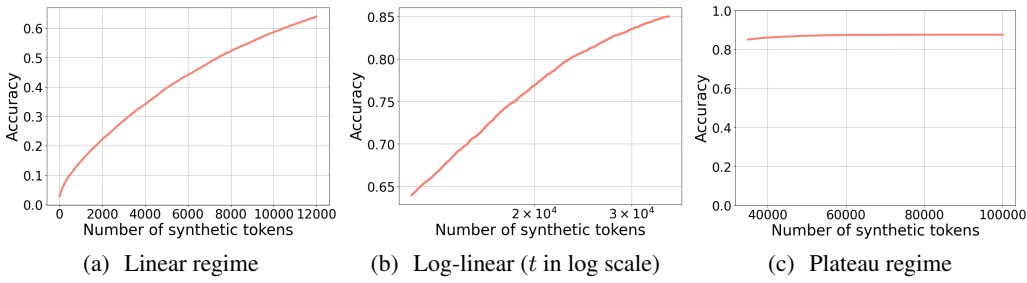

(a) Linear regime        (b) Log-linear ($t$ in log scale)        (c) Plateau regime

Figure 9: Accuracy $\text{Acc}(M_t)$ with respect to time $t$, for $V = 100$ and $p = 0.03$. The mixture-of-exponential functional form in (2) leads to three distinct regimes.

**Qualitative description.** Finally, to help build an intuitive understanding, we provide a qualitative description of the mixture-of-exponential shape. We demonstrate in Appendix G that this mixture-of-exponential shape comprises three distinct phases: a fast growth phase, a slower growth phase, and a plateau phase. Mathematically, we show the existence of two distinct times, $0 < t_1 < t_2$, such that

$$\text{Acc}(M_T) = \begin{cases} \Theta\left(p + t\right), & \text{for } 0 \leq t \leq t_1, \\ \Theta(\log t), & \text{for } t_1 \leq t \leq t_2, \\ \Theta(1), & \text{for } t \geq t_2, \end{cases}$$

where we use a convenient change of variable $T = tV(V - 1)$. It is important to note that the choice of $\log t$ in the second phase is not necessarily canonical. In fact, the bound holds for any well-behaved monotone increasing concave function as a replacement for $\log t$. Our representation here is motivated by two factors: first, it aligns with the performance observed in our EntiGraph CPT numerical results, and second, it reflects the gradual slowdown in growth. We illustrate the three phases in Figure 9, which present a simulation of the toy model with $p = 0.03$.

To perform curve fitting using the mixture-of-exponential formula, we approximate the infinite sum with three terms in

$$\mathsf{Acc}(\boldsymbol{M}_t) \sim p + C \left( 1 - \sum_{k=1}^{\infty} \mu(k) \left( 1 - a_k \right)^t \right).$$

Mathematically, we fit the empirical observation against the formula

$$y(x) = a - b_1 r_1^x - b_2 r_2^x - b_3 r_3^x,$$

where $x$ is the EntiGraph token count (in millions) and $y(x)$ is the QuALITY QA accuracy. We use the non-linear least squares method implemented by Virtanen et al. (2020). As a result of this procedure, we obtain the fitted formula

$$y(x) = 64.5456 - 13.8352 \times (0.9989)^x - 8.4705 \times (0.8961)^x - 3.932 \times (0.0546)^x.$$

For the implementation of this procedure, we refer readers to our code release.

# H  SYNTHETIC DATA GENERATION PROMPTS

We generate two synthetic corpora in this paper: EntiGraph (Appendix H.1) and the Rephrase baseline (Appendix H.2). In our experiments, the $\mathcal{D}_{\text{source}}$ is a collection of documents $D$, and our synthetic augmentation procedure is applied to each document $D \in \mathcal{D}_{\text{source}}$. We will focus on a single document $D$ for the remainder of this section.

## H.1  ENTIGRAPH PROMPTS

The EntiGraph procedure is described in detail in §2.2. We will recap the three steps below.

**Step 1: Entity extraction.**  The first step is to extract the salient entities from the document $D$ using the `entity_extraction` operation (Step 1, §2.2). The complete `entity_extraction` prompt is as follows:

```
As a knowledge analyzer, your task is to dissect and understand an
article provided by the user. You are required to perform the
following steps:
1. Summarize the Article: Provide a concise summary of the entire
article, capturing the main points and themes.
2. Extract Entities: Identify and list all significant "nouns" or
entities mentioned within the article. These entities should include
but not limited to:
    * People: Any individuals mentioned in the article, using the
    names or references provided.
    * Places: Both specific locations and abstract spaces relevant to
    the content.
    * Object: Any concrete object that is referenced by the provided
    content.
    * Concepts: Any significant abstract ideas or themes that are
    central to the article's discussion.

Try to exhaust as many entities as possible. Your response should be
structured in a JSON format to organize the information effectively.
Ensure that the summary is brief yet comprehensive, and the list of
entities is detailed and accurate.

Here is the format you should use for your response:

{
  "summary":  "<A concise summary of the article>",
  "entities": ["entity1", "entity2", ...]
}
```

**Step 2: Relation analysis.**  The last step is to generate diverse descriptions of relations among two or more entities. In our experiments, for each document $D$, we enumerate all entity pairs and generate a description for each. The prompt for generating a description relating a pair of entities is as follows:

```
You will act as a knowledge analyzer tasked with dissecting an
article provided by the user. Your role involves two main
objectives:
1. Rephrasing Content: The user will identify two specific entities
   mentioned in the article. You are required to rephrase the
   content of the article twice:
    * Once, emphasizing the first entity.
    * Again, emphasizing the second entity.
2. Analyzing Interactions: Discuss how the two specified entities
   interact within the context of the article.
```

```
Your responses should provide clear segregation between the
rephrased content and the interaction analysis. Ensure each section
of the output include sufficient context, ideally referencing the
article's title to maintain clarity about the discussion's focus.
Here is the format you should follow for your response:

### Discussion of <title> in relation to <entity1>
<Rephrased content focusing on the first entity>

### Discussion of <title> in relation to <entity2>
<Rephrased content focusing on the second entity>

### Discussion of Interaction between <entity1> and <entity2>
    in context of <title>
<Discussion on how the two entities interact within the article>
```

We also generate synthetic data involving three entities, using the prompt below:

```
You will act as a knowledge analyzer tasked with dissecting an
article provided by the user. Your role involves three main
objectives:

1. Rephrasing Content: The user will identify three specific
   entities mentioned in the article. You are required to rephrase
   the content of the article three times:
    * Once, emphasizing the first entity.
    * Again, emphasizing the second entity.
    * Lastly, emphasizing the third entity.
2. Analyzing Interactions: Discuss how these three specified
   entities interact within the context of the article.

Your responses should provide clear segregation between the
rephrased content and the interaction analysis. Ensure each section
of the output include sufficient context, ideally referencing the
article's title to maintain clarity about the discussion's focus.
Here is the format you should follow for your response:

### Discussion of <title> in relation to <entity1>
<Rephrased content focusing on the first entity>

### Discussion of <title> in relation to <entity2>
<Rephrased content focusing on the second entity>

### Discussion of <title> in relation to <entity3>
<Rephrased content focusing on the third entity>

### Discussion of Interaction between <entity1>, <entity2> and
    <entity3> in context of <title>
<Discussion on how the three entities interact within the article>
```

## H.2 REPHRASE PROMPTS

For the rephrase corpus, we adapt the prompt from Maini et al. (2024) to our setting of books and articles. We provide four rephrase styles below:

**Easy rephrase:**

```
You are an assistant to help read a article and then rephrase it in
simpler terms. The user will provide you with an article with
```

```
title, year, content. You need to generate a paraphrase of the same
article using a very small vocabulary and extremely simple
sentences that a toddler will understand. Remember to keep the
meaning and every content of the article intact, including the
title, year, etc.
```

**Medium rephrase:**

```
You are an assistant to help read a article and then rephrase it in
different terms. The user will provide you with an article with
title, year, content. You need to generate a paraphrase of the same
article using diverse and high quality English language as in
sentences on Wikipedia. Remember to keep the meaning and every
content of the article intact, including the title, year,
etc.
```

**Hard rephrase:**

```
You are an assistant to help read a article and then rephrase it in
more sophisticated terms. The user will provide you with an article
with title, year, content. You need to generate a paraphrase of the
same article using very terse and abstruse language that only an
erudite scholar will understand. Remember to keep the meaning and
every content of the article intact, including the title, year,
etc.
```

# I ADDITIONAL EVALUATION DETAILS OF MAIN EXPERIMENTS

## I.1 QuALITY QA QUESTION SET

In this section, we provide more details of evaluation on the QuALITY QA test queries. Throughout the closed-book QA experiments, we use a fixed 5-shot prompt below:

```
## Example 1
### Question
In the context of "Les Misérables", written by Victor Hugo in 1862,
what is the main setting of the novel? There is only one correct
choice.
### Choices
A. London
B. Madrid
C. Paris
D. Rome
### Thought Process and Answer
Thought process: "Les Misérables" is primarily set in Paris, making
C the correct choice. London, Madrid, and Rome are significant
cities in other literary works but not in Victor Hugo's "Les
Misérables". There is only one correct choice.
Answer: C.

## Example 2
### Question
In the context of "Brave New World", written by Aldous Huxley in
1932, what substance is widely used in the society to control
citizens' happiness? There is only one correct choice.
### Choices
A. Gold
B. Soma
C. Silver
D. Iron
### Thought Process and Answer
Thought process: In Aldous Huxley's "Brave New World," Soma is used
as a means to maintain social control by ensuring citizens'
happiness, making B the correct choice. Gold, Silver, and Iron are
not the substances used for this purpose in the book.
Answer: B.

## Example 3
### Question
In the context of "Romeo and Juliet", written by William
Shakespeare in the early 1590s, what are the names of the two
feuding families? There is only one correct choice.
Choices:
A. Montague and Capulet
B. Bennet and Darcy
C. Linton and Earnshaw
D. Bloom and Dedalus
### Thought Process and Answer
Thought process: In William Shakespeare's "Romeo and Juliet," the
two feuding families are the Montagues and the Capulets, making A
the correct choice. The Bennets and Darcys are in "Pride and
Prejudice", the Lintons and Earnshaws in "Wuthering Heights", and
Bloom and Dedalus in "Ulysses".
Answer: A.

## Example 4
### Question
```

```
In the context of "1984", written by George Orwell in 1949, what is
the name of the totalitarian leader? There is only one correct
choice.
### Choices
A. Big Brother
B. O'Brien
C. Winston Smith
D. Emmanuel Goldstein
### Thought Process and Answer
Thought process: In George Orwell's "1984," the totalitarian leader
is known as Big Brother, making A the correct choice. O'Brien is a
character in the novel, Winston Smith is the protagonist, and
Emmanuel Goldstein is a rebel leader.
Answer: A.

## Example 5
### Question
In the context of "Moby-Dick", written by Herman Melville in 1851,
what is the name of the ship's captain obsessed with hunting the
titular whale? There is only one correct choice.
### Choices
A. Captain Hook
B. Captain Nemo
C. Captain Flint
D. Captain Ahab
### Thought Process and Answer
Thought process: In Herman Melville's "Moby-Dick," the ship's
captain obsessed with hunting the whale is Captain Ahab, making D
the correct choice. Captain Nemo is in "Twenty Thousand Leagues
Under the Sea", Captain Flint in "Treasure Island", and Captain
Hook in "Peter Pan".
Answer: D.

## Example 6
```

If the output of the model correctly follows the format of the few-shot prompt, its last two characters should be "`A.`", "`B.`", "`C.`", or "`D.`". However, the model sometimes cannot successfully follow the few-shot prompting format, particularly for the continually pretrained model. As a result, in all our evaluations, we sample the response 64 times, and only select the ones that can be parsed in the correct format. Out of these 64 attempts, we randomly select among the valid answers to give the final answer. Note that this is *different* from majority voting in self-consistency prompting (Wang et al., 2023a).

## I.2 CLOSED-BOOK SUMMARIZATION

**Automated evaluation metric.** We design a three-stage evaluation procedure: (i) In the first stage, we use GPT-4[5] to break the summary into atomic claims, similar to Min et al. (2023); (ii) In the second stage, we provide both the list of claims and the source article to a judge model (also GPT-4). We ask the judge model to determine whether each claim is true or false, based on the source article. If the claim is true, we further ask the model to determine whether the claim is salient (contributes to the main message of the article) or cosmetic (factual details that do not help understand the main message). (iii) Finally, for each summary, we obtain its number of false and salient claims and normalize it by the corresponding count from the human summary. We report the average of these normalized metrics across the QuALITY corpus articles in Figure 3.

**Prompts to generate summaries.** For summarization evaluation with EntiGraph Instruct and Raw Instruct, we apply the following two prompts to obtain two summaries of increasing length. We provide three examples of summarization outputs below. For each of the three examples, we will

---

[5]Specifically, we use the `gpt-4-turbo` model as of Aug. 19, 2024.

➤ **Short prompt:** `Summarize the article {article title} by {author name} for me.`

`Give a short summary of ``Cosmic Yo-Yo'' by Ross Rocklynne.`

➤ **Long prompt:** `Write an extremely long and detailed article regarding the book {article title} by {author name}.`

`Write an extremely long and detailed article regarding the book ``Cosmic Yo-Yo'' by Ross Rocklynne.`

Table 5: Summarization prompt for EntiGraph Instruct, Raw Instruct, and Reprhase Instruct.

first present the human summary for this article to provide context for the example, and then present the short summary from the two summarizers.

**Example 1.** The first example is "Cosmic Yo-Yo" by Ross Rocklynne.

> **Human summary:** Bob Parker, the President of Interplanetary Hauling & Moving Co., sells asteroids to wealthy people on earth. Clients ask for asteroids with size parameters and specifications, and Bob finds them in space and hauls them to earth. His company is almost bankrupt because a rival company, Saylor & Saylor, stole his idea and now offers the same services. Bob receives mail from Mr. Andrew S. Burnside with a request for an asteroid that he would like to use in an upcoming wedding. Bob and his partner Queazy set out to find the perfect asteroid for Mr. Burnside, although they know it's a longshot. Fairly quickly, they find one that looks perfect. The men land on the asteroid, and Bob deploys his atomic-whirl spectroscope to test it. Suddenly, a beautiful woman interrupts him and demands that they leave the asteroid. She pulls out her spasticizer gun before telling them that they can have it in a month after she's gone. Bob explains that they are desperate, but the girl retorts that her fate is worse than death if she leaves.
> Suddenly, the Saylor brothers' ship appears, and Bob tells the girl that they have to fight this enemy together. Wally and Billy Saylor, along with three other men, jump out of the ship. Bob tells them that Mr. Burnside has ordered this asteroid, and the Saylor brothers say that they received the same order. Bob quickly grabs the girl's spasticizer while Queazy throws his body at Billy. However, Wally manages to shoot the gun out of Bob's hand and attack him. Bob is knocked unconscious in the scuffle.
> When Bob wakes up, he is completely alone, floating in space. He panics because he has very little oxygen left. Finally, he hears Queazy's voice explaining that the girl used her ship's technology to find them both. The mystery girl introduces herself as Starre Lowenthal, the granddaughter of Mr. Burnside. She concedes that this entire mission was fake. She told her grandfather that she would only marry her fiance Mac if he could get this particular asteroid, and then she made plans to conquer and protect the asteroid so it could not be supplied for the wedding.
> Bob is confident that they can reach the Saylor brothers before they bring the asteroid back to earth, but his plan does nothing to protect Starre from marrying a man she doesn't love. She agrees to help Bob and Queazy. Within five days, Bob realizes he is in love with Starre. Starre compares her small ship to a yo-yo, and Bob gets an idea - they will use Starre's ship like a yo-yo to retrieve the asteroid from the Saylor brothers. Once the team catches up to the Saylor brothers, Bob flings Starre's ship at the asteroid several times, and Wally calls them to tell them that they might die as a result of the damage their ship has sustained. Bob makes it clear that they have no intention of stopping, and the Saylor brothers release the asteroid.

> **EntiGraph Instruct with short prompt:** Cosmic Yo-Yo is a science fiction story by Ross Rocklynne about Bob Parker and Queazy, two interplanetary haulers who are hired to deliver an asteroid to Earth for a wealthy client's backyard wedding. However, they face competition from their rivals, the Saylor brothers, who also want to deliver the asteroid. The story follows their race to deliver the asteroid before the deadline, using a yo-yo-like device to maneuver

the asteroid and outsmart their rivals. The story is filled with action, adventure, and a touch of romance.

**Raw Instruct with short prompt:** Cosmic Yo-Yo by Rocklynne, Ross is a science fiction novel that follows the story of a young woman named Susan who discovers a mysterious device that can transport her through time and space. She uses the device to travel to different planets and meet different alien species, but soon realizes that her actions have consequences and she must find a way to return home. The novel explores themes of identity, responsibility, and the consequences of our actions.

**Example 2.** The second example is "Beach Scene" by Marshall King. We can see that EntiGraph correctly gives a summary of the story while Raw Instruct mistakes the book as a painting.

**Human summary:** Purnie, an animal, is going to see the ocean on his fifth birthday. He has heard stories about this place, and experiencing it firsthand is surreal for him. Purnie is careful not to disturb the animals he sees along the way because he has frozen time, and everything must resume normally when he unfreezes it. He knows that time-stopping is forbidden for animals his age, but he chooses to believe that his family will be proud of his bravery.

Finally, he sees the ocean in front of him, and he resumes time. He does a head-stand and feels weak and dizzy. These feelings are a result of the time-stop, and he knows it. Purnie approaches some humans on the beach. A man named Forbes is in the middle of explaining to his captain, Benson, that he has found 17 planets to claim as his own. Forbes is hellbent on raising his FORBES flag as soon as possible. He is eager to stake his claim to the land and says that his mission is much bigger than real estate alone. Benson retorts that yes, his mission is bigger than just real estate because his paperwork says that Forbes will own all of the inhabitants of the planets he claims as well as the land.

The crew members use a special machine and find radiation emanating from Purnie. Forbes demands that they put the animal in a box. Benson protests and reminds Forbes that it's against Universal Law, but Forbes insists. Purnie experiences his first-ever impulse to run away with fear when a noose comes towards him. He goes back to pick up his fruit, and Forbes shoots him in the leg. When the man throws the noose again, Purnie involuntarily stops time. He drags himself up the knoll where he originally came from.

The humans are astonished when time resumes and Purnie is not where he was a split second ago. They spot him up on top of a pile of petrified logs, and suddenly the logs fall down the hill and pin the men down. Purnie is shocked and regretful. The whole thing was an accident. He deliberately stops time and uses all of his remaining strength to lift the logs off of the humans. Purnie begins to lose consciousness, and he knows that he must resume time or he will die. After pouring all of his strength into this action, time does begin again. The humans resume life and feel as though they have gone mad. They know that they were just facing death by drowning, and now they are free. The logs were so heavy that it would have taken superhuman strength to move them. Forbes, in particular, has really gone mad, and he laughs to himself uncontrollably. Benson believes that Purnie was responsible for moving the logs, but of course that seems physically impossible.

Purnie stares off at the beautiful ocean views and watches the men leave in their vehicle as he dies.

**EntiGraph Instruct with short prompt:** Beach Scene is a short story by Marshall King that follows a young boy named Purnie who discovers the power of stopping time. On his fifth birthday, Purnie decides to explore the beach and use his newfound ability to stop time to play with the animals and explore the ocean. However, his adventures take a turn when he encounters a group of explorers who want to capture him for their own gain. Purnie must use his powers to save himself and the animals from the explorers' greed.

> **Raw Instruct with short prompt:** Beach Scene by King, Marshall is a painting that depicts a group of people enjoying a day at the beach. The painting features a vibrant blue ocean, a sandy beach, and a group of people relaxing, playing, and swimming in the water. The painting is known for its bright colors and lively atmosphere, capturing the joy and energy of a day at the beach.

**Example 3.** The third example is "The Madman From Earth" by "Keith Laumer".

> **Human summary:** Following the departure of Consul Whaffle, Retief has taken over as Consul for the Terrestrial States with the Terrestrial Consulate General on the planet Groac. His administrative assistant, Miss Yolanda Meuhl, wants him to attend Groacian cultural events, but Retief is more interested in addressing the nine-year-old mystery of the disappearance of a Terrestrial cruiser, the ISV Terrific–an event which was followed by a coup d'etat enacted by the current Groacian government. Much to Miss Meuhl's dismay, Retief shirks his cultural duties and makes his way to the Foreign Office Archives, whereupon he is promptly barred from entering by a pale-featured Archivist speaking in the throat-bladder vibrations of the native Groacians. Because of the Archivist's insistence that öutworldersc̈annot access the archives, Retief begins walking back to the Consulate and stops at a bar for a drink. At the, a drunken Groacian approaches Retief and threatens to cage him and put him on display as a freak. The bartender orders the drunken Groacian out of the bar, and Retief follows him, ultimately beating him up for information. When Retief returns to the Consulate, Miss Meuhl informs him that two angry Groaci await him in his office. One is Fith, an employee of the Terrestrial Desk at the Ministry of Foreign Affairs; the other is Shluh, a representative of the Internal Police. They are there to investigate reports that Retief has assaulted a Groacian national–an accusation Retief ignores in order to launch into his own accusations that the Groaci were engaged in a cover-up of the whereabouts of the ISV Terrific. Miss Meuhl continually interjects, expresses horror at Retief's claims, and apologizes to the Groacians on behalf of the Terrestrial Consulate. Despite the persistent denials of the Groacians, Retief continues his accusations, suggesting the coup d'etat was an effort to silence government officials with knowledge of the truth of what happened to the cruiser and its crew. Then he reveals what he discovered from the drunken Groacian: The crew of the ISV Terrific had been caged and paraded through the streets of Groac and then neglected until they died. Fith and Shluh finally admit the truth and offer to show Retief the hidden cruiser in order to show their contrition. When Retief sees the ship, he once again accuses the Groacians of attempting to mislead him, saying that this is a lifeboat, and he demands to see the actual ship. Fith has had enough and threatens to arrest Retief, who yields and goes back to the Consulate. There, Miss Meuhl is at her wits end. Retief orders her to barricade herself inside the office while he goes to the Foreign Ministry to gather more evidence. When he returns, Miss Meuhl informs him she has reported him to Regional Headquarters, and Retief learns he has been relieved of his post. Soon after, the Groacians appoint Miss Meuhl to his position, and Fith and Shluh enter to arrest him.

> **EntiGraph Instruct with short prompt:** The Madman From Earth is a science fiction novel by Keith Laumer, published in 1959. The story follows Retief, a diplomat from Earth, who is sent to the planet Groac to investigate the disappearance of a Terrestrial cruiser and its crew. Retief uncovers a conspiracy by the Groacian government to cover up the incident and hide evidence from Earth. He must navigate through political intrigue, cultural misunderstandings, and dangerous situations to uncover the truth and bring justice to the missing crew. The novel is a satirical take on diplomacy and bureaucracy, with Retief's unorthodox methods and wit standing out against the rigid and often absurd procedures of the diplomatic corps.

> **Raw Instruct with short prompt:** The Madman From Earth by Laumer, Keith is a science

fiction novel that follows the story of a man named John Smith who is transported to a distant planet called Earth Two. There, he discovers that he is the only human on the planet and must navigate a world filled with strange creatures and dangerous obstacles. Despite his initial confusion and fear, John eventually adapts to his new surroundings and becomes a respected member of the alien society. However, when he is eventually transported back to Earth One, he realizes that he has been gone for over a thousand years and must grapple with the changes that have occurred during his absence.

➤ **Explicit article reference instruction:** Summarize the article "Defining Decay Down" by "David Plotz" for me.

The article "Defining Decay Down" by David Plotz discusses how the dental industry has changed over the years. In the past, dentists were primarily focused on treating cavities and other dental issues, but as the prevalence of cavities decreased due to the widespread use of fluoride, the industry had to adapt. Dentists began to focus on cosmetic dentistry, offering services like teeth whitening and orthodontics to appeal to the vanity of aging baby boomers. The article also discusses how dentists have become more customer-friendly, with amenities like virtual reality glasses and massage therapy. Despite the shift towards elective procedures, the dental industry has continued to thrive, with dentists earning higher salaries than ever before.

➤ **Implicit article reference:** How has the dentistry practice in United States has changed?

The dentistry practice in the United States has undergone significant changes in recent years. Here are some of the ways in which it has changed:
1. Focus on cosmetic dentistry: In recent years, there has been a growing demand for cosmetic dental procedures such as teeth whitening, orthodontics, and veneers. Dentists have responded to this demand by offering more cosmetic services.
2. Use of technology: Dentists are now using advanced technology such as intra-oral cameras, 3D imaging, and laser technology to diagnose and treat dental problems. This technology has made dental procedures more accurate and efficient.
3. Emphasis on preventive care: Dentists are now focusing more on preventive care to help patients avoid dental problems. This includes regular check-ups, cleanings, and fluoride treatments.
4. Increased competition: With the rise of dental schools and the influx of foreign-trained dentists, the dental industry has become more competitive. Dentists are now offering more services and amenities to attract and retain patients.
5. Focus on patient experience: Dentists are now focusing on providing a positive patient experience. This includes offering amenities such as virtual reality glasses, massage therapy, and entertainment during procedures.
Overall, the dentistry practice in the United States has become more patient-centered, technology-driven, and focused on preventive care and cosmetic procedures.

➤ **Cross article instruction:** Compare David Plotz's commentary style between how he analyze American dentistry and how he discuss the movie Fight Club?

David Plotz's commentary style is different when he analyzes American dentistry and when he discusses the movie Fight Club. In his analysis of American dentistry, Plotz uses a critical and investigative approach, highlighting the over-commercialization and unnecessary procedures in the industry. He uses statistics and expert opinions to support his arguments and provides practical advice for readers to avoid unnecessary dental procedures.
On the other hand, when discussing the movie Fight Club, Plotz adopts a more satirical and humorous tone. He uses pop culture references and witty commentary to analyze the themes and messages of the movie. He also provides a critical analysis of the film's characters and their motivations, while also acknowledging the movie's cultural impact.
Overall, Plotz's commentary style is more serious and investigative when analyzing American dentistry, while his commentary on Fight Club is more satirical and humorous.

Table 6: Complete instruction following example used in Table 2 from Section 4.3.

