# OpenReview forum: "Synthetic continued pretraining"
_ICLR.cc/2025/Conference — ICLR 2025 Oral_

### Official Review · Reviewer_nNRV · 2024-10-28

**Soundness:** 4
**Presentation:** 3
**Contribution:** 3
**Rating:** 8
**Confidence:** 3

**Summary:**

This paper addresses how to train LLMs in a data scarce regime, given that LLMs require O(1000) examples of a fact to actually "learn" it. This has applications both to niche corpora (e.g., mathematics textbooks) as well as to training larger models once all human text is exhausted. The authors propose to use a pre-trained LLM to (1) extract entities and summaries from a comparatively small, niche corpus, and (2) use the extracted entities to generate rephrased assertions about those entities, to facilitate learning by a second (here, smaller) LLM. They experiment with a 1.3M token reading comprehension dataset, and test the approach against several baselines, including closed-book tests on the LLM used to extract entities and the rephrased text used to train the second LLM. Finally, the authors present a mathematical model through which they attempt to understand the training behavior of this data augmentation system.

**Strengths:**

The experiments are convincing that the EntiGraph approach improves the LLM's ability to accurately answer questions about a small corpus. In particular the closed-book results in Figure 3 show that the EntiGraph approach leads to far more salient claims per false claim than any of the other models, including GPT-4, or training the LLM (Llama 3 8B). The benefit is substantially less in the open-book RAG case, but there is still substantial improvement. The theoretical model to explain how the model improves QA accuracy with increasing tokens provides some good intuition as to how the model learns.

Overall the text is clear and easy to read.

**Weaknesses:**

I still have reservations that there is some amount of distillation of GPT-4 into their Llama 3 8B: it seems possible to me that a RAG-prompted GPT4 could generate additional information that is somehow "unlocked" by the RAG prompt, but which the closed-book version was unable to access. At the risk of anthropomorphizing, this is akin to a human getting a visual or audio cue and suddenly recalling whole complex memories. It would make the paper stronger to dig into the results of entity extraction and the generated text to see whether it is rephrasing/paraphrasing, or whether possibly actual new information is injected.

Even so, it would have helped this reader to have pointed out the significance of the closed book experiments earlier on. It isn't stated explicitly until the Limitations section.

I don't feel particularly qualified to check your proofs of theorems, and moreover I think the main value of the theoretical model is to help the reader understand intuitively why the approach works (these may be connected observations). Is all of the theory necessary? Perhaps a simulation would do as well?

Another issue is that much of the benefit of the approach vanishes (though not completely) when using a RAG model directly. Is this approach worth the extra training, given the modest gains? The core problem, really, is how many examples LLMs take to learn anything well. This paper finds a way to side-step that successfully, but doesn't solve it directly.

**Questions:**

The paper could be more robust if you had more than just the QuALITY dataset. It is a perennial problem to find hard datasets to work with, so I understand this may be all there is for now, but given the chance I would attempt to reproduce the results on a different set. The authors mention linear algebra (a much harder topic, I think): is there any corpus for that subject?

The presentation of how exactly you generate the text to train Llama 3 8B with EntiGraph is still a little fuzzy to me, in particular it would be nice to see some examples of what you generated. It is helpful to have the prompts, but some output always grounds the presentation.

Finally, I imagine GPT-4t made errors in producing the training data--did you search for these? Even at a quick glance how often did it make errors, and what, if anything, did you do to filter them out?

---

> ### Author Response · Authors · 2024-11-21
>
> Dear Reviewer nNRV,
>
> Thanks for your hard work and helpful feedback! Below we address your specific comments as best as we can, and we hope you will engage with us during the discussion period to clarify any remaining points.
>
> ## Risk of Distillation from GPT-4 Data Generator
> Thank you for pointing out the risk of distillation with GPT-4. To more concretely test this, we have performed an experiment with a weaker Llama 3.1 8B Instruct generator, and discuss our results in the general comment.
>
> ## Additional Datasets on Harder Topics
>
> Thanks for suggesting that we test out Synthetic CPT on a harder dataset. We have conducted an experiment using lecture transcripts and the Coursera Exam QA dataset and discuss these results in the general comment.
>
> ## Hallucination in Synthetic Data
>
> Thanks for noting that we could see hallucinations with our synthetic data generator. We have conducted a human evaluation to test the factuality of EntiGraph-generated data, which we discuss in the general comment. We did not explicitly filter out hallucinations in the paper’s experiments, and ultimately find in the human evaluation that the hallucination rate is very low.
>
> ## Role of Theory Section
>
> The role of the theory section is to explain why Synthetic CPT does not need to create new knowledge de novo to improve performance. The core intuition is explained at the beginning of the section in L417-L421. Regarding the mathematical complexity, the main mathematical result is actually a simple generalization of the celebrated coupon collector’s problem with a minor twist: instead of collecting new coupons, EntiGraph collects new knowledge, but the probability of collecting a particular piece of new knowledge differs. As a result, instead of having an exponential growth as in the coupon collector’s problem, we end up with a mixture of exponential growth.
>
> We hope that our response and additional experiments have adequately addressed your concerns about distillation, further evaluation on harder datasets, and hallucination in the synthetic data.
>
> We would greatly appreciate it if you could engage with us during the discussion period on any remaining barriers to raising your score.

---

> ### Comment · Reviewer_nNRV · 2024-11-25
>
> The additional experiments are quite helpful, I've changed my scores accordingly.

---

### Official Review · Reviewer_wsFi · 2024-10-31

**Soundness:** 2
**Presentation:** 3
**Contribution:** 2
**Rating:** 8
**Confidence:** 4

**Summary:**

The paper proposes "synthetic continued pretraining" to enhance language models' domain adaptation efficiency, particularly when adapting to a small corpus of domain-specific documents. The authors introduce EntiGraph, a synthetic data augmentation algorithm that constructs a knowledge graph from the entities in the original corpus and synthesizes diverse text to enhance the pretraining process. The approach was evaluated using a reading comprehension dataset, QuALITY, showing notable improvements in model performance with both closed-book and open-book scenarios.

**Strengths:**

1. The proposed EntiGraph approach for generating synthetic data is well-motivated and demonstrates clear improvements in downstream performance, as shown by the experimental results.
2. The paper includes comprehensive evaluations, including closed-book QA, instruction following, and open-book settings. The results show a significant performance improvement over baselines, validating the effectiveness of synthetic pretraining.
3. The authors provide a theoretical analysis of EntiGraph's effectiveness, which aligns well with empirical observations and provides a deeper understanding of its scaling properties.

**Weaknesses:**

1. The evaluation relies on the QuALITY dataset, which may not be representative of all types of small corpora. A broader range of datasets, particularly from diverse domains, would make the results more generalizable.
2. Although the authors attempt to mitigate hallucinations by grounding synthetic data generation in the original corpus, the risk of generating inaccurate information is inherent in using a language model for synthetic generation. This aspect needs further empirical examination, such as quantitative metrics to evaluate hallucination rates.
3. The approach relies on using strong language models like GPT-4 for synthetic data generation. The practical feasibility of using this approach might be limited if users do not have access to such models due to their computational cost. What if it was replaced with LLama 3 8B?
4. While the paper includes useful baselines such as "Rephrase CPT," more comparisons with alternative data augmentation or synthetic generation methods from recent literature could strengthen the claim that EntiGraph is an effective strategy.

**Questions:**

1. How sensitive is the synthetic pretraining process to the specific hyperparameters used for entity extraction and relation analysis? Would tuning these parameters significantly affect the generated corpus quality?

2. How does the synthetic corpus compare to a manually curated dataset in terms of quality and impact on downstream tasks?

3. Could EntiGraph be used effectively in scenarios where entities are ambiguous or domain-specific (e.g., medical or legal texts)?

---

> ### Author Response · Authors · 2024-11-21
>
> Dear Reviewer wsFi,
>
> Thanks for your hard work and helpful feedback! Below, we address your specific comments as best as we can, and we hope you will engage with us during the discussion period to clarify any remaining points.
>
> ## Evaluation with Other Datasets from Diverse Domains
>
> Thank you for your suggestion to show that Synthetic CPT works on other datasets and domains. We have conducted an experiment using lecture transcripts and the Coursera Exam QA dataset and discuss these results in the general comment.
>
> ## Quantitative Evaluation of Hallucination Rates in Generated Text
>
> Thanks for suggesting that we quantitatively measure the hallucination rate of the generated text. We have measured the factuality and lexical diversity of our synthetic corpora and discuss these results in the general comment.
>
> ## Demonstrating Synthetic CPT with an Open-Source Synthetic Data Generator
>
> We agree that demonstrating Synthetic CPT works with a weaker, open-source synthetic data generator would be useful. We have performed this experiment and discuss our results in the general comment.
>
> ## Alternative Data Augmentations
>
> To the best of our knowledge, our work is the first to propose an augmentation specifically designed for learning in a data-constrained setting. Therefore, there are not many established baselines, leading us to adapt the prompts of WRAP [1] to our setting. We believe it is exciting future work to design and benchmark new augmentations for this setting. However, given the cost of end-to-end experiments, we believe this is out-of-scope for this paper.
>
> ## Comparison to Manually Curating Data
>
> We are specifically interested in learning from corpora with niche knowledge, such as rare articles/books or cutting-edge technical content. This knowledge is by construction not well-represented in standard internet text. Moreover, some of the reading comprehension questions in our test set refer directly to the source document (“what did X say about the topic”), and therefore the baseline of Raw CPT on source documents could be viewed as a form of strong data curation baseline.
>
> ## Ambiguous or Domain-Specific Entities
>
> We believe that our existing experiments with QuALITY already demonstrate that EntiGraph works well with domain-specific entities, because QuALITY is composed of niche narratives with unique characters, objects, or concepts (e.g., entities such as “ethergram”, “Satterfield City”, and “spasticizer”). Moreover, our experiments with Coursera more directly test EntiGraph’s effectiveness with real-world, specialized entities from technical content.
>
> Thank you again for your thorough suggestions to help improve the paper. We hope that our response has adequately addressed your concerns, such as generalization to other datasets, hallucination in the synthetic data, and use of a strong data generator.
>
> We would greatly appreciate it if you could engage with us during the discussion period on any remaining barriers to raising your score.
>
> [1] Maini et al, 2024. Rephrasing the Web: A Recipe for Compute and Data-Efficient Language Modeling. https://arxiv.org/abs/2401.16380

---

> > ### Comment · Reviewer_wsFi · 2024-11-26
> >
> > Thank you for your clarification and additional experiments. I will raise the score.

---

### Official Review · Reviewer_ipvv · 2024-11-02

**Soundness:** 4
**Presentation:** 3
**Contribution:** 3
**Rating:** 8
**Confidence:** 3

**Summary:**

This paper proposes a method to continue pretraining LLM with a synthetic data augmentation method. The method is based on expanding the training corpus with many verbalizations of the entity graph present in the training corpus. It moves from a sparsely verbalized entity graph to a more densily verbalized one by using only the source documents and prompting LLMs to generate the new tokens.

The paper shows that the method is beneficial for downstream tasks in closed- and open-book QA as well as RAG.

Overall, I think the paper is worthy of acceptance, it propose a clean method with good results and the experiments are fairly convincing.

**Strengths:**

The paper does a good job at demonstrating the benefit of the synthetically generated data, by including relevant natural baselines.
The proposed method seem to work well and can be useful for continued pre-training tasks.

**Weaknesses:**

The work relies on commercial and closed-source models (GPT4) for generating the synthetic data, making this work non-reproducible. Since the data generation process is the central contribution, it would have been interesting to have insights about how well different models can perform this data generation task.

The paper proposes only extrinsic evaluation of the generated data but does not provide intrinsic measures, i.e., how good is the generated text?

In my opinion, section 6 is not particularly useful. It is unnecessarily mathematical, based on simplistic assumptions and does not bring useful insights (For many continuously increasing lines, there anyway exists a mixture-of-exponential that fit it)

**Questions:**

For the data generator, what type of models are necessary to have good performance? (why use GPT4 and not open-source models)
The paper shows that the generated data is useful, but how does it look like? (is it good quality text, factual, natural looking, ...)
What is the significance of section 6?

---

> ### Author Response · Authors · 2024-11-21
>
> Dear Reviewer ipvv,
>
> Thanks for your hard work and helpful feedback! Below we address your specific comments as best as we can, and we hope you will engage with us during the discussion period to clarify any remaining points.
>
> ## Demonstrating Synthetic CPT with an Open-Source Synthetic Data Generator
>
> Thank you for your suggestion to investigate whether Synthetic CPT works with a weaker, open-source synthetic data generator. We have performed this experiment and discuss our results in the general comment.
>
> ## Intrinsic Evaluation of Generated Text
>
> Thanks for suggesting that we provide intrinsic measures of the generated text. We have measured the factuality and lexical diversity of our synthetic corpora and discuss these results in the general comment.
>
> ## Role of Theory Section
> The role of the theory section is to explain why Synthetic CPT does not need to create new knowledge de novo to improve performance. The core intuition is explained at the beginning of the section in L417-L421. Regarding the mathematical complexity, the main mathematical result is actually a simple generalization of the celebrated coupon collector’s problem with a minor twist: instead of collecting new coupons, EntiGraph collects new knowledge, but the probability of collecting a particular piece of new knowledge differs. As a result, instead of having an exponential growth as in the coupon collector’s problem, we end up with a mixture of exponential growth.
>
> We hope that our response, experiments with a weaker generator, and intrinsic evaluation of the generated data have adequately addressed your concerns.

---

> > ### Comment · Reviewer_ipvv · 2024-11-25
> > **Thanks for the answer**
> >
> > Thank you for the clarification and answers, I find the rebuttal experiments convincing and I'll keep my 'accept' score.

---

### Official Review · Reviewer_9zM9 · 2024-11-03

**Soundness:** 4
**Presentation:** 4
**Contribution:** 3
**Rating:** 8
**Confidence:** 4

**Summary:**

This paper addresses the problem of data inefficiency in pretraining language models. Current pretraining corpora may not generalize effectively and models may benefit from structured, repeated, diverse representations of knowledge.

The proposed is a two-step process that (1) extracts entities from the corpus and then (2) extracts relationship information amongst a subset of the entities.

Experimentation uses the QuALITY corpus and dataset, which is a benchmark for long-document reading comprehension. Evaluation compares with relevant baselines like training on the original QuALITY corpus and a corpus containing rephrasings.

**Strengths:**

* The problem the work addresses is important.
* Experimental results show that this method scales better than simple paraphrasing or direct pretraining, and that retrieval-augmented generation further boosts performance of this model.
* The authors also present a theoretical model explaining EntiGraph’s log-linear scaling pattern, providing insights into the mechanics of synthetic data’s impact on learning efficiency.
* Paper is clear and well-written.

**Weaknesses:**

While the experiments focus on the QuALITY corpus, it remains unclear how well this would apply to other domain-specific corpora or more complex fields (e.g., legal or math data).

**Questions:**

It says “We generate data for pairs D_{Ei, Ej} and triplets D_{Ei, Ej, Ek} in our experiments”. I wonder if the authors have any intuition about how performance changes with the size of subset k.

---

> ### Author Response · Authors · 2024-11-21
>
> Dear Reviewer 9zM9,
>
> Thanks for your hard work and helpful feedback\! Below we address your specific comments as best as we can, and we hope you will engage with us during the discussion period to clarify any remaining points.
>
> ## Experiments on Other Domain-Specific Corpora
>
> Thank you for your suggestion of additional evaluation on domain-specific corpora in a more  complex field. We have conducted an experiment using lecture transcripts and the Coursera Exam QA dataset and discuss these results in the general comment.
>
> We hope that our response has adequately addressed your concerns about generalization of Synthetic CPT to other domains.

---

> > ### Comment · Reviewer_9zM9 · 2024-11-27
> >
> > Thanks! Will keep my positive score.

---

### Author Response · Authors · 2024-11-21
**Response to all reviewers**

We thank the reviewers for their hard work and detailed feedback.

We were glad that you support our motivation, mentioning that “the problem… is important” (9zM9) and that the EntiGraph approach is “well-motivated” (wsFi). Further, we are happy you found that our method is “clean” (ipvv) and that the theoretical model “provides insights into the mechanics of synthetic \[data\]” (9zM9), “aligns well with empirical observations and provides a deeper understanding of \[EntiGraph’s\] scaling properties” (wsFi), and “provides some good intuition” (nNRV).

We were also glad to see you highlight our empirical evaluation, mentioning that the experiments/evaluations are “fairly convincing” (ipvv), “convincing” (nNRV), and “comprehensive” (wsFi), and the downstream performance improvements are “notable”, “clear”, and “significant” (wsFi). Lastly, we are excited that the paper was “clear and well-written” (9zM9) and “clear and easy to read” (nNRV).

Based on your feedback, we **conducted three additional experiments to better understand EntiGraph** and its properties. **Full results, setup details, and discussion are in Appendix I** (last 3 pages) of the manuscript, and we will add pointers in the main text. We summarize the results below:

## Rebuttal Experiment 1: Ablation with weaker synthetic data generator (Appendix I.1)
First, based on the feedback of ipvv, wsFi, and nNRV, we conducted experiments **replacing the strong GPT-4-Turbo-based synthetic data generator with a weaker Llama 3.1 8B Instruct model**. This accounts for the concern that the gains of synthetic CPT arise from distilling a stronger model.

We observe consistent gains with the Llama 3.1 8B generator up to 334M synthetic tokens. Moreover, the slope of the scaling trend is similar to EntiGraph’s trend using the GPT-4-Turbo generator. In contrast, the log-linear slope is smaller for the Rephrase baseline (with Llama 3.1 8B Instruct, the same weaker synthetic data generator) as in our main scaling results in Figure 2 (Section 4.2). Altogether, this experiment demonstrates that **the gains of Synthetic CPT with EntiGraph are significant and reproducible even using standard open-source models**.

## Rebuttal Experiment 2: Factuality and lexical diversity of generated synthetic data (Appendix I.2)
Second, reviewers ipvv, wsFi, and nNRV suggested we obtain quantitative, intrinsic measures of our synthesized corpora. We conducted a **human evaluation of the factuality** of the EntiGraph corpus, and **measured lexical diversity by computing n-gram overlap statistics** between EntiGraph- and Rephrase-generated text and the source documents.

Factuality: we have paper authors annotate random sentences from EntiGraph and find that **94.4% of non-subjective sentences are factually supported by the source document**. Contrasting with GPT-4’s poor closed-book performance on QuALITY (\~51%), this experiment shows that grounding with source documents helps prevent hallucination.

Lexical diversity: for both EntiGraph and Rephrase-generated text, we find that only a small percentage of n-grams are exactly copied from the source documents. For example, less than \~0.5% of 8-grams and \~0.2% of 16-grams in the EntiGraph and Rephrase synthetic corpora appear in the source documents. This suggests that both methods produce lexically diverse knowledge representations.

## Rebuttal Experiment 3: New dataset beyond QuALITY  (Appendix I.3)
Lastly, reviewers 9zM9, wsFi, and nNRV suggest that we **test our method on another domain**, ideally in a “complex field” (9zM9) or a “harder topic” (nNRV). To that end, we **conducted scaling experiments using the Coursera Exam QA dataset \[1\]**, which contains 15 lecture transcripts and exam questions from advanced technical courses. **This** **setting is even more data-scarce, with only 124K raw tokens** (compared to 1.3M in QuALITY). We find that EntiGraph consistently delivers log-linear improvement in accuracy, outperforming the base model and Rephrase baseline.

**Altogether, these carefully constructed evaluations show that Synthetic CPT robustly enables an LM to learn from niche domains, and that these gains arise from diverse representations of knowledge rather than distillation.**

Thanks again for the helpful feedback. We hope to engage with you to clarify any remaining points\!

\[1\] An et al., 2023\. L-Eval: Instituting Standardized Evaluation for Long Context Language Models.

---

### Meta-Review · Area_Chair_gbTc · 2024-12-19

**Metareview:**

The paper proposes a data-synthesizing method for continued pretraining for adapting an LLM to a specific domain, where the proposed approach is based on EntiGraph including entities and their relations. In addition, the authors derived bounds for their scaling trends.

All reviewers agree that this is an interesting and solid paper. Reviewers have raised a few minor concerns and questions, which may be addressed in the revision.

**Additional Comments On Reviewer Discussion:**

Reviewers unanimously recommended acceptance.

---

### Decision · Program_Chairs · 2025-01-22

Accept (Oral)